# Joint neutrino oscillation analysis from the T2K and NOvA experiments

The NOvA Collaboration*✉ & The T2K Collaboration*✉

The landmark discovery that neutrinos have mass and can change type (or flavour) as they propagate—a process called neutrino oscillation[1-6]—has opened up a rich array of theoretical and experimental questions being actively pursued today. Neutrino oscillation remains the most powerful experimental tool for addressing many of these questions, including whether neutrinos violate charge-parity (CP) symmetry, which has possible connections to the unexplained preponderance of matter over antimatter in the Universe[7-11]. Oscillation measurements also probe the mass-squared differences between the different neutrino mass states ($\Delta m^2$), whether there are two light states and a heavier one (normal ordering) or vice versa (inverted ordering), and the structure of neutrino mass and flavour mixing[12]. Here we carry out the first joint analysis of datasets from NOvA[13] and T2K[14], the two currently operating long-baseline neutrino oscillation experiments (hundreds of kilometres of neutrino travel distance), taking advantage of our complementary experimental designs and setting new constraints on several neutrino sector parameters. This analysis provides new precision on the $\Delta m^2_{32}$ mass difference, finding $2.43^{+0.04}_{-0.03} \times 10^{-3}$ eV$^2$ in the normal ordering and $-2.48^{+0.03}_{-0.04} \times 10^{-3}$ eV$^2$ in the inverted ordering, as well as a $3\sigma$ interval on $\delta_{CP}$ of $[-1.38\pi, 0.30\pi]$ in the normal ordering and $[-0.92\pi, -0.04\pi]$ in the inverted ordering. The data show no strong preference for either mass ordering, but notably, if inverted ordering were assumed true within the three-flavour mixing model, then our results would provide evidence of CP symmetry violation in the lepton sector.

The standard model of particle physics, extended to include neutrino mass, describes three-flavour eigenstates of neutrinos ($\nu_e$, $\nu_\mu$, $\nu_\tau$) that are related to three mass eigenstates ($\nu_1$, $\nu_2$, $\nu_3$) by the 3 × 3 complex Pontecorvo–Maki–Nakagawa–Sakata unitary mixing matrix $U_{PMNS}$ (refs. 15–17). This mixing, together with non-zero neutrino mass, allows for the phenomenon of neutrino oscillation, in which, during propagation, the flavour content of a neutrino evolves at a rate that depends on neutrino mass-squared splittings ($\Delta m^2_{ij} \equiv m_i^2 - m_j^2$) and the $U_{PMNS}$ matrix elements. Apart from these oscillation parameters, the rate depends on neutrino energy $E_\nu$ and neutrino propagation distance $L$ (baseline). Although experiments studying this process in recent decades have provided insights into the details of neutrino masses and mixings[12], many open questions remain.

The mixing matrix $U_{PMNS}$ is typically parameterized in terms of three mixing angles ($\theta_{12}$, $\theta_{13}$, $\theta_{23}$) and at least one complex phase $\delta_{CP}$ (ref. 12). It is unknown whether $\sin \delta_{CP}$ is non-zero; if it is, neutrinos—and thus leptons—violate charge-parity (CP) symmetry and thereby provide a source of matter–antimatter asymmetry in nature[17], which is of great interest given the connection between CP violation and the unexplained matter dominance in the Universe[7-11]. Separately, oscillation experiments have established that the mass-squared splitting $\Delta m^2_{32}$ is roughly 30 times larger in magnitude than $\Delta m^2_{21}$, but the sign of the former is at present unknown. That is, $\nu_3$ may be heavier or lighter than the $\nu_1$–$\nu_2$ pair, with these two options termed, respectively, the normal ($\Delta m^2_{32} > 0$) and inverted ($\Delta m^2_{32} < 0$) mass orderings. Knowledge of the mass

ordering can constrain experimental searches and theory development in a wide range of physics, including absolute neutrino mass measurements[18], neutrinoless double beta decay searches to investigate the nature of neutrino mass[19], models of supernova explosion and detection[20,21], and the cosmological evolution evidenced in cosmic microwave background and large-scale structure measurements[22]. For the mixing angles, current data suggest $\theta_{23}$ is near 45°, a notable value hinting at a $\mu/\tau$ flavour symmetry[17]. Improved precision on this and other mixing angles is essential for gaining a clearer view of flavour mixing and to probe the validity of the three-flavour model.

Long-baseline accelerator neutrino oscillation experiments are well suited to address the above questions. In these, a high-intensity neutrino beam enriched in muon neutrinos ($\nu_\mu$) or muon antineutrinos ($\bar{\nu}_\mu$) is produced at a particle accelerator and directed through the crust of Earth towards a massive far detector located hundreds of kilometres away. Note that the word 'neutrino' is used to mean both neutrino and antineutrino unless stated otherwise. The far detector measures the event rates of $\nu_\mu$ and $\nu_e$—the latter primarily from $\nu_\mu \to \nu_e$ oscillation—as a function of neutrino energy, from which the oscillation parameters above can be determined. These experiments use near detectors, sited a short distance from the beam source, in which oscillation effects are negligible and a very high neutrino event rate can be measured. The near detectors provide vital control measurements that substantially mitigate large systematic uncertainties in the initial neutrino flux, neutrino-on-nucleus interaction cross-sections and in some cases

---

*List of authors and their affiliations appears at the end of the paper. ✉e-mail: nova-spokespersons@fnal.gov; t2k-spokesperson@mailman.t2k.org

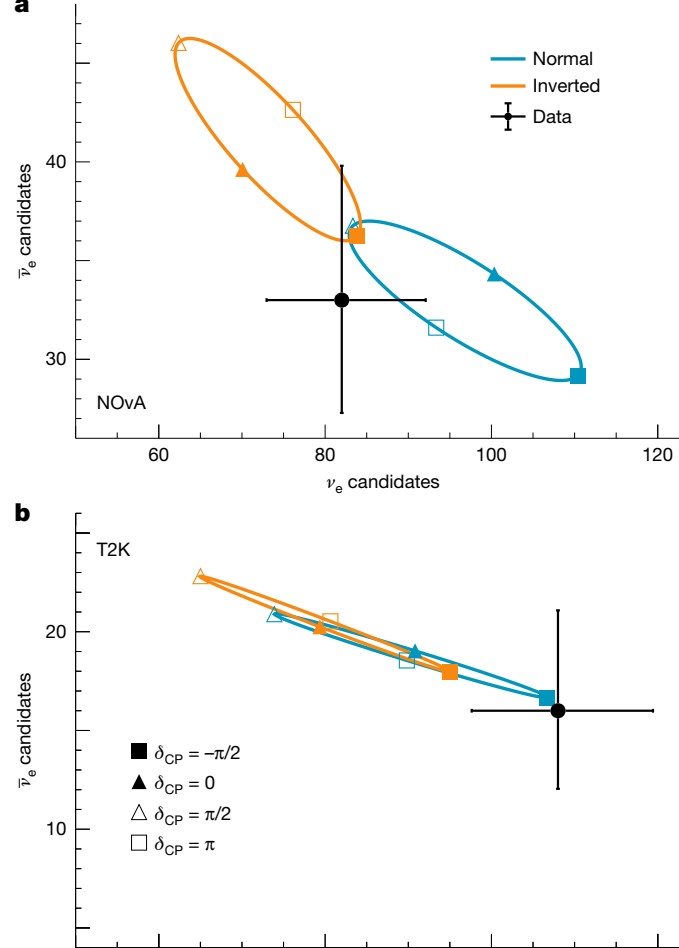

**a**

Normal
Inverted
Data

$\bar\nu_e$ candidates

NOvA

$\nu_e$ candidates

**b**

T2K

$\bar\nu_e$ candidates

$\delta_{CP} = -\pi/2$
$\delta_{CP} = 0$
$\delta_{CP} = \pi/2$
$\delta_{CP} = \pi$

$\nu_e$ candidates

**Fig. 1 | The impact of mass ordering and $\delta_{CP}$ on event rates. a,b,** A bi-event plot that shows experimental sensitivity to neutrino mass ordering and $\delta_{CP}$, with panels representing the NOvA (**a**) and T2K (**b**) cases. Black points with $1\sigma$ Poisson statistical error bars show the total number of $\nu_e$ and $\bar\nu_e$ candidates selected in the far detectors. The oval parametric curves trace out predicted numbers of events under the normal (blue) or inverted (orange) mass ordering assumption as the parameter $\delta_{CP}$ varies from $-\pi$ to $\pi$. Four specific $\delta_{CP}$ values are labelled for reference. All other oscillation parameters are kept fixed in this graphic, set to their most probable values from the joint analysis (Extended Data Table 3).

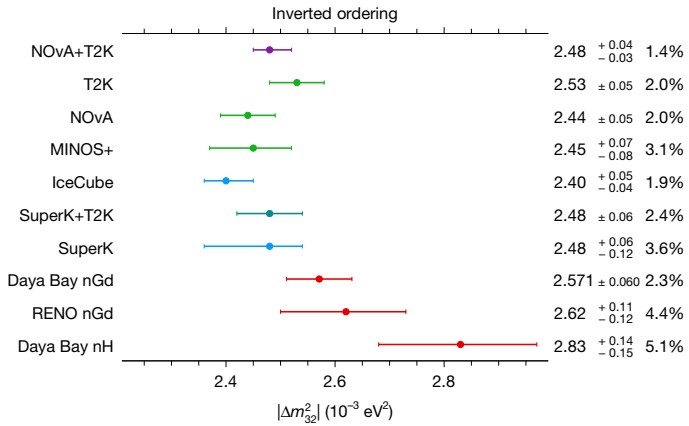

Inverted ordering

| | | | |
|---|---|---|---|
| NOvA+T2K | | $2.48 \, ^{+0.04}_{-0.03}$ | 1.4% |
| T2K | | $2.53 \pm 0.05$ | 2.0% |
| NOvA | | $2.44 \pm 0.05$ | 2.0% |
| MINOS+ | | $2.45 \, ^{+0.07}_{-0.08}$ | 3.1% |
| IceCube | | $2.40 \, ^{+0.05}_{-0.04}$ | 1.9% |
| SuperK+T2K | | $2.48 \pm 0.06$ | 2.4% |
| SuperK | | $2.48 \, ^{+0.06}_{-0.12}$ | 3.6% |
| Daya Bay nGd | | $2.571 \pm 0.060$ | 2.3% |
| RENO nGd | | $2.62 \, ^{+0.11}_{-0.12}$ | 4.4% |
| Daya Bay nH | | $2.83 \, ^{+0.14}_{-0.15}$ | 5.1% |

$|\Delta m^2_{32}| \, (10^{-3} \, eV^2)$

**Fig. 2 | Experimental measurements of $|\Delta m^2_{32}|$.** The measurements assume the inverted ordering preferred by this analysis. Sources for the results from top to bottom, starting with the second line, are as follows: refs. 13,14,43–49. The normal ordering case is available in Extended Data Fig. 9.

the $\nu_\mu \to \nu_e$ oscillation probability is a function of (among other things) both $\delta_{CP}$ and the neutrino mass ordering, and these two effects must be teased apart.

Figure 1 shows the complementarity between the experiments in a simplified case. Sets of oval curves indicate the energy-integrated total $\nu_e$ and $\bar\nu_e$ event counts expected in the far detectors under various mass ordering and $\delta_{CP}$ scenarios, with other oscillation parameters held fixed. The measured event counts in NOvA and T2K are shown as black points with error bars.

As shown in Fig. 1a, there is stronger separation between the mass ordering ovals for NOvA, because of higher beam energies, but as the NOvA data lie near the overlap of the ellipses, there can be ambiguity as to which ordering is correct and (in a correlated way) which values of $\delta_{CP}$ are preferred. By contrast, T2K has less sensitivity to the mass ordering, but points with similar values of $\delta_{CP}$ in each hierarchy sit close to one another, and the data lie closest to $\delta_{CP} = -\frac{\pi}{2}$, regardless of mass ordering. Combining these datasets can provide simultaneous mass ordering and $\delta_{CP}$ information with substantially less ambiguity, maximizing the use of current data and informing data-taking strategies for current and future experiments.

This discussion points to a more general observation that the oscillation parameters of interest represent a highly correlated multidimensional space. The analysis reported here calculates a joint Bayesian posterior, using the likelihoods of the experiments defined over the full parameter space. Moreover, we use the full suite of analysis tools from both experiments: detector response models, neutrino energy estimators, near-detector measurements and systematic uncertainties, all within a unified framework for statistical inference. This level of integration is the first for accelerator neutrino experiments, to our knowledge.

The posterior calculation is based on detailed parameterized models of the neutrino flux, cross-sections and detectors that predict the binned spectra of neutrino events in each of our selected samples as a function of the oscillation parameters and a large number of nuisance parameters mostly related to systematic uncertainties in the models. A likelihood is constructed from Poisson probability terms describing the compatibility between the prediction and the observed data in bins of relevant variables. Prior probabilities are set on all parameters as detailed in the Methods.

Both T2K and NOvA have software that explores the posterior using Markov chain Monte Carlo (MCMC) methods[28,29] (ARIA for NOvA[30] and MaCh3 for T2K[31]). By containerizing[32] the likelihood and prior portions of the code, we can construct and analyse the joint posterior using either of the original MCMC frameworks, in spite of the very different

detector response (for example, energy reconstruction and event selection efficiencies).

Two such experiments are in operation today, T2K and NOvA. Each experiment uses a narrow-band off-axis beam[23,24], whose peak energy is near the first oscillation maximum, $\sin^2\left(\frac{\Delta m^2_{32}L}{4E}\right) \approx 1$, at the far detector. Note that natural units, where $\hbar = c = 1$, are used throughout. T2K uses an approximately 0.6 GeV neutrino beam from J-PARC in Tokai, Japan, and the 50-kt Super-Kamiokande water Cherenkov detector for its far detector located 295 km away[25]. In the United States, an approximately 2 GeV beam of NOvA is produced at Fermilab near Chicago, and the 14-kt tracking calorimeter far detector is located 810 km away in northern Minnesota[26]. Further details on the designs of NOvA and T2K and on long-baseline experiments can be found in the Methods and refs. 25–27.

We report here a combined analysis of the datasets from T2K and NOvA previously analysed independently in refs. 13,14. This combination takes advantage of marked complementarity in the sensitivities of the two experiments to the oscillation parameters. In particular,

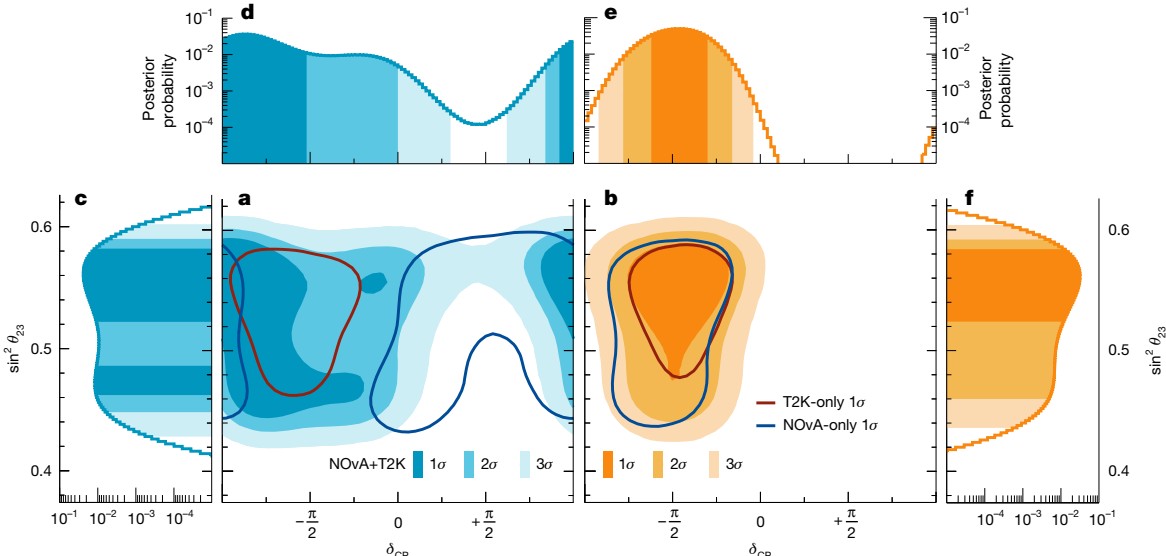

**Fig. 3 | Constraints on $\sin^2\theta_{23}$ and $\delta_{CP}$.** Marginalized posterior probabilities and 1D or 2D Bayesian credible regions of $\sin^2\theta_{23}$ and $\delta_{CP}$ in the case of the normal (blue, left side) and inverted (orange, right side) neutrino mass ordering with the reactor constraint applied. Shaded areas correspond to $1\sigma$, $2\sigma$ and $3\sigma$ credible regions. **a,b,** The 2D panels of $\sin^2\theta_{23}$ vs $\delta_{CP}$ (**a,b**) are overlaid with $1\sigma$ credible regions from the T2K-only (dark red) and NOvA-only (dark blue) data fits assuming normal (**a**) and inverted ordering (**b**). **c–f,** The 1D panels show the posterior probabilities of $\sin^2\theta_{23}$ (**c**) and $\delta_{CP}$ (**d**) in the normal ordering, and $\delta_{CP}$ (**e**) and $\sin^2\theta_{23}$ (**f**) in the inverted ordering.

software environments involved. For each fitting framework, ARIA or MaCh3, the native likelihood and priors of the fitter are calculated directly, whereas the likelihood and priors of other experiments are accessed using the software container. In this way, either framework can be used, providing valuable redundancy and thus cross-checks of all statistical inferences.

Although a single set of oscillation parameters naturally applies to both experiments in the joint posterior, the treatment of the many nuisance parameters related to systematic uncertainties is more subtle. Both measurements of the oscillation parameters at present have statistical uncertainties larger than the systematic uncertainties, but the latter are not negligible. We thoroughly surveyed the flux, cross-section and detector models and their systematic uncertainties to determine whether correlations between the experiments affect the analysis at a significant level. Our conclusions from this effort are summarized in the following paragraphs.

Both T2K and NOvA use beams produced by directing accelerated protons onto graphite targets. The hadrons are charge-selected with magnetic horns: positively charged hadrons decay to produce neutrinos, and negatively charged hadrons produce antineutrinos. Many uncertainties on these beam fluxes stem from processes unrelated between the two experiments, for example, the alignment of beam components. Yet, uncertainties on the rate of hadron production in the graphite targets are substantial, and the underlying physics is the same. However, the two experiments use proton beams of different energies (30 GeV for T2K and 120 GeV for NOvA), and the external datasets used to tune the hadron production models of both experiments are different[33–35]. Moreover, the ultimate impact of flux uncertainties on far-detector predictions in NOvA is much smaller than other uncertainties. We, therefore, conclude that at current experimental exposures, the flux uncertainties of the two experiments need not be correlated.

Given the different detector technologies involved, most detector-related uncertainties are independent between the experiments. Furthermore, the very different energy estimation techniques used, combined with model tuning and uncertainty estimation using in situ calibration samples in each experiment, including for the lepton and neutron energy scales, lead to independence between the two detector

uncertainty models. We conclude that there are no significant correlations in the detector models.

For neutrino-on-nucleus cross-sections, the underlying physics is the same; in many cases, the same external datasets are used by both experiments to tune and set prior uncertainties on model parameters. Thus, cross-section model correlations are expected. However, in the specific case of NOvA and T2K, the description of this physics differs markedly. The simulation packages differ[36,37], the physical models implemented in them differ in many places, the parameterizations differ almost entirely, and customized tunings are necessary and applied, given the specific energies of the experiments, detector technologies and approaches to systematic uncertainty mitigation.

Proper correlations between experiments could be implemented by starting from a common cross-section model spanning different energy ranges and able to describe both the leptonic and hadronic parts of the final state. A joint description is not yet mature and is one of the focuses of the community in the years to come[38]. Given the differences in the models, a direct mapping of their parameters was deemed not practical at this time. Instead, we studied whether neglecting these correlations could appreciably affect our measurements of the oscillation parameters. The studies are limited to our current experimental exposures and models and would need re-evaluation if applied to any other context.

First, we assessed whether correlations between single systematic parameters in our models could have a substantial impact on our results. For each of $\Delta m_{32}^2$, $\theta_{23}$ and $\delta_{CP}$, we identified the systematic parameter in each experiment with the largest impact on the measurement of that oscillation parameter. Then, regardless of whether those two systematic parameters made physical sense to correlate, we performed fits to simulated pseudo-data with the parameters fully correlated, uncorrelated and fully anticorrelated. Details of these studies, including how we identified the most impactful parameters, are shown in the Methods. In summary, we saw no case in which the choice of correlation of individual systematic parameters significantly affected the oscillation parameter measurements.

Checking individual parameters does not rule out effects from a mix of systematic parameter variations that combine to produce a net effect that is larger and possibly more degenerate with oscillation

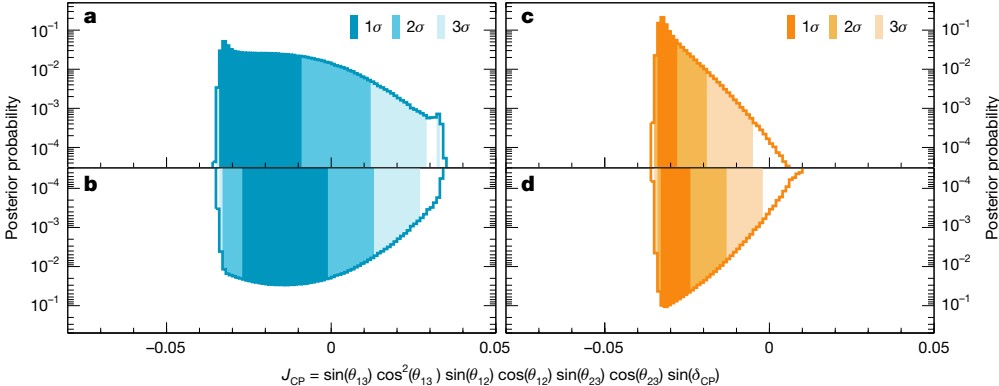

**Fig. 4 | Constraints on the Jarlskog invariant. a–d,** Marginalized posterior probabilities of the Jarlskog invariant, $J_{CP}$, in the case of the normal (blue; **a,b**) and inverted (orange; **c,d**) neutrino mass ordering with the reactor constraint applied. The posterior distributions use prior distributions either flat in $\delta_{CP}$ (**a,c**) or sin $\delta_{CP}$ (**b,d**). Shaded areas show the 1$\sigma$, 2$\sigma$ and 3$\sigma$ Bayesian credible intervals.

effects, representing a potential worst-case scenario for the analyses. Rather than seeking such a set of variations, we directly identified, or in some cases constructed, single systematic parameters for each experiment that have effects similar to each oscillation parameter of interest. We then adjusted the size of the priors on these 'nightmare' parameters such that their impact on the measurements is comparable to that of statistical errors and therefore larger than the net effect of all our regular systematic parameters combined. These nightmare parameters were added to our nominal uncertainty models to create augmented models, allowing us to study a case in which systematic effects are comparable to statistical uncertainty. Next, we constructed simulated pseudo-datasets with the nightmare parameters increased in both experiments by one standard deviation above their prior central values. These simulated pseudo-data were then fit three times using the augmented model: once with the nightmare parameters of the experiments fully correlated (matching the pseudo-data), once fully anticorrelated and finally uncorrelated. We find that the oscillation parameter constraints extracted in the fully correlated and uncorrelated cases have negligible differences. However, the incorrect anticorrelated case yields a large bias. We expect that with even larger systematic uncertainties, differences between the correlated and uncorrelated cases would eventually become relevant. However, this study indicates that we are not in such a regime with the current exposures and systematic uncertainties (see the Methods for further results).

Given that no significant biases are seen from neglecting correlations between actual systematic parameters, and the only bias seen with the nightmare parameters comes not from neglecting a correlation but from adding an incorrect one, we choose in most cases to neglect the correlations between the systematic uncertainties of the two experiments. The one exception relates to the approximately 2% normalization uncertainties on all $\nu_e$ and $\bar{\nu}_e$ events described in ref. 39. In this case, the uncertainties are implemented identically by T2K and NOvA, and we have correlated them.

We also perform studies in which the joint fit is tested against pseudo-data constructed with a set of discrete model variations not directly accessible using the nominal uncertainty models of the experiments. This procedure was used in the earlier independent T2K analysis[14], and we include in the present analysis those model variations seen as most impactful previously. Similarly, we studied a secondary set of variations based on extrapolating the cross-section model of each experiment to the context of the other experiment. Predefined thresholds were used to establish that no substantive changes in the central values or interval widths of the oscillation parameters were seen under these tests, as described in the Methods. For all tested alternative models, all observed changes in credible intervals were within thresholds (see the Methods for further details). Each experiment continues to investigate

improvements in its cross-section models, and the studies described here would warrant repeating for larger data exposures and/or updated theoretical understanding. Continued theoretical and experimental effort in this direction is important.

With the joint likelihood and systematic uncertainty model defined, we use our fitting frameworks to analyse the combined datasets of refs. 13,14, finding consistent results between the two frameworks. Unless stated otherwise, we report results using an external constraint on $\theta_{13}$ (named the 'reactor constraint' below) and external constraints on $\Delta m^2_{21}$ and $\theta_{12}$. The values used for these constraints correspond to the 2020 Particle Data Group summary values[40] and are given in the Methods.

We tested the goodness of fit (Methods) of our model to data using the $P$-value method[41], both overall and for each individual sample in the far detectors. All the $P$-values are within an acceptable range (>0.05 after the look-elsewhere-effect adjustment described in the Methods). The overall $P$-value to describe all NOvA and T2K samples is 0.75 for full spectral analysis and 0.40 for rate-only analysis, marginalized over both mass orderings. Similar results were obtained without the reactor constraint and in each mass ordering. Thus, the joint oscillation model simultaneously fits T2K and NOvA data well. The $P$-values are also consistent with those of previous T2K-only and NOvA-only analyses.

We produce parameter estimations using the highest-posterior-density credible intervals and perform discrete hypothesis tests using the Bayes factor formalism. Conclusions related to CP conservation or violation, $\Delta m^2_{32}$, $\sin^2\theta_{23}$ and mass ordering have been tested to be robust under the alternative model variations described previously. For the measured oscillation parameters, we report 1$\sigma$ (68.27%) credible intervals unless noted.

We find $\sin^2\theta_{23} = 0.56^{+0.03}_{-0.05}$ without any assumptions on the ordering of the neutrino masses. The fit weakly prefers the upper octant of $\theta_{23}$ ($\sin^2\theta_{23} > 0.5$) over the lower octant with a Bayes factor of 3.5. Removing the reactor constraint gives no statistically significant preference for either octant (Bayes factor 1.2 for the lower octant compared with the upper octant). We also find $\Delta m^2_{32} = 2.43^{+0.04}_{-0.03} \times 10^{-3}$ eV$^2$ assuming the normal ordering and $\Delta m^2_{32} = -2.48^{+0.03}_{-0.04} \times 10^{-3}$ eV$^2$ assuming the inverted ordering. This is at present the smallest experimental uncertainty on $|\Delta m^2_{32}|$ (Fig. 2), to our knowledge. This conclusion also applies when the reactor constraint is replaced by a flat prior.

There is no statistically significant preference obtained for either of the mass orderings, with a Bayes factor of 1.3 in favour of the inverted ordering with reactor $\theta_{13}$ constraint and 2.5 without reactor $\theta_{13}$ constraint. Although the two experiments individually prefer the normal ordering, the values of other oscillation parameters are more consistent in the inverted ordering, leading to a different ordering preference in the joint fit, although still not statistically significant. The effect on

mass ordering preference when additionally incorporating reactor $\Delta m_{32}^2$ measurements is discussed in the Methods.

With no assumption on the true mass ordering, we find the $1\sigma$ credible interval on $\delta_{CP}$ to contain $[-0.81\pi, -0.26\pi]$ with the highest posterior probability value being $-0.47\pi$. We also find that values of $\delta_{CP}$ around $+\pi/2$, an extremum of $\sin\delta_{CP}$, are outside our $3\sigma$ (99.73%) credible intervals, which also holds for either mass ordering separately. Figure 3 shows the joint fit result compared with the individual measurements of NOvA and T2K in the $\sin^2\theta_{23} - \delta_{CP}$ plane, as well as one-dimensional (1D) uniformly binned posterior probability distributions for both mass ordering cases. Assuming the normal ordering, the joint analysis allows a wide range of $\delta_{CP}$ values, giving a $3\sigma$ credible interval of $\delta_{CP} \in [-1.38\pi, 0.30\pi]$. In the case of the inverted ordering $\delta_{CP} \in [-0.92\pi, -0.04\pi]$, excluding 56% of the parameter space, the CP-conserving values of $\delta_{CP} = 0$ and $\pi$ are outside the $3\sigma$ credible interval. A consistent picture is seen when analysing the Jarlskog invariant, $J_{CP}$ (ref. 42), which is a parametrization-independent measure of CP violation. The CP-conserving value of $J_{CP} = 0$ falls outside the $3\sigma$ credible interval for the inverted ordering, and the above statements are true whether the prior used is uniform in $\delta_{CP}$ or $\sin\delta_{CP}$ (Fig. 4). This analysis, therefore, provides evidence for CP violation in the lepton sector if the inverted ordering is assumed to be true. However, we do not see a significant preference at present for either mass ordering. Future mass ordering measurements will, therefore, influence the interpretation of these results. See the Methods for more data projections and comparisons.

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

## The NOvA Collaboration

S. Abubakar[1], M. A. Acero[2], B. Acharya[3], P. Adamson[4], N. Anfimov[5], A. Antoshkin[5], E. Arrieta-Diaz[6], L. Asquith[7], A. Aurisano[8], D. Azevedo[9], A. Back[10,11], N. Balashov[5], P. Baldi[12], B. A. Bambah[13], E. F. Bannister[7], A. Barros[2], A. Bat[1,14], K. Bays[15], R. Bernstein[4], T. J. C. Bezerra[7], V. Bhatnagar[16], B. Bhuyan[17], J. Bian[12,15], A. C. Booth[7,18], R. Bowles[10], B. Brahma[19], C. Bromberg[20], N. Buchanan[21], A. Butkevich[22], S. Calvez[21], J. M. Carceller[23], T. J. Carroll[24,25],

E. Catano-Mur[26], J. P. Cesar[24], R. Chirco[27], B. C. Choudhary[28], A. Christensen[21], M. F. Cicala[23], T. E. Coan[29], T. Contreras[4], A. Cooleybeck[30], D. Coveyou[30], L. Cremonesi[18], G. S. Davies[3], P. F. Derwent[4], P. Ding[4], Z. Djurcic[31], K. Dobbs[32], M. Dolce[33], D. Dueñas Tonguino[8], E. C. Dukes[30], A. Dye[3], R. Ehrlich[30], E. Ewart[10], P. Filip[34], M. J. Frank[35], H. R. Gallagher[36], A. Giri[19], R. A. Gomes[9], M. C. Goodman[31], R. Group[30], A. Habig[37], F. Hakl[38], J. Hartnell[7], R. Hatcher[4], J. M. Hays[18], M. He[32], K. Heller[15], V. Hewes[8], A. Himmel[4], T. Horoho[30], A. Ivanova[5], B. Jargowsky[12], I. Kakorin[5], A. Kalitkina[5], D. M. Kaplan[27], A. Khanam[39], B. Kirezli[1], J. Kleykamp[3], O. Klimov[5], L. W. Koerner[32], L. Kolupaeva[5], R. Kralik[7], A. Kumar[16], C. D. Kuruppu[40], V. Kus[41], T. Lackey[4,10], K. Lang[24], L. Lasorak[7], J. Lesmeister[32], A. Lister[25], J. Liu[12], J. A. Lock[7], M. MacMahon[23], S. Magill[31], W. A. Mann[36], M. T. Manoharan[42], M. Manrique Plata[10], M. L. Marshak[15], M. Martinez-Casales[4,11], V. Matveev[22], B. Mehta[16], M. D. Messier[10], H. Meyer[33], T. Miao[4], W. H. Miller[15], S. R. Mishra[40], R. Mohanta[13], A. Moren[37], A. Morozova[5], W. Mu[4], L. Mualem[43], M. Muether[33], K. Mulder[23], D. Myers[24], D. Naples[44], S. Nelleri[42], J. K. Nelson[26], R. Nichol[23], E. Niner[4], A. Norman[4], A. Norrick[4], H. Oh[8], A. Olshevskiy[5], T. Olson[32], M. Ozkaynak[23], A. Pal[45], J. Paley[4], L. Panda[45], R. B. Patterson[43], G. Pawloski[15], R. Petti[40], R. K. Plunkett[4], J. C. C. Porter[7], L. R. Prais[3,8], A. Rafique[31], V. Raj[43], M. Rajaoalisoa[8], B. Ramson[4], B. Rebel[25], E. Robles[12], P. Roy[33], O. Samoylov[5], M. C. Sanchez[11,46], S. Sánchez Falero[11], P. Shanahan[4], P. Sharma[16], A. Sheshukov[5], Shivam[17], A. Shmakov[12], W. Shorrock[7], S. Shukla[47], I. Singh[28], P. Singh[18,28], V. Singh[47], S. Singh Chhibra[18], D. K. Singha[13], A. Smith[15], J. Smolik[41], N. Solomey[33], A. Sousa[8], K. Soustruznik[48], M. Strait[4,15], L. Suter[4], A. Sutton[11,46], S. Swain[45], C. Sweeney[23], A. Sztuc[23], N. Talukdar[40], P. Tas[48], T. Thakore[8], J. Thomas[23], E. Tiras[1,11], M. Titus[42], Y. Torun[27], D. Tran[32], J. Trokan-Tenorio[26], J. Urheim[10], P. Vahle[26], Z. Vallari[49], K. J. Vockerodt[18], A. V. Waldron[18], M. Wallbank[4,8], T. K. Warburton[11], C. Weber[15], M. Wetstein[11], D. Whittington[10,39], D. A. Wickremasinghe[4], J. Wolcott[36], S. Wu[15], W. Wu[12], W. Wu[44], Y. Xiao[12], B. Yaeggy[8], A. Yahaya[33], A. Yankelevich[12], K. Yonehara[4], S. Zadorozhnyy[22], J. Zalesak[34] & R. Zwaska[4]

[1]Department of Physics, Erciyes University, Kayseri, Turkey. [2]Universidad del Atlantico, Puerto Colombia, Colombia. [3]University of Mississippi, Lafayette, MS, USA. [4]Fermi National Accelerator Laboratory, Batavia, IL, USA. [5]Joint Institute for Nuclear Research, Dubna, Russia. [6]Universidad del Magdalena, Santa Marta, Colombia. [7]Department of Physics and Astronomy, University of Sussex, Brighton, UK. [8]Department of Physics, University of Cincinnati, Cincinnati, OH, USA. [9]Instituto de Física, Universidade Federal de Goiás, Goiânia, Brazil. [10]Indiana University, Bloomington, IN, USA. [11]Department of Physics and Astronomy, Iowa State University, Ames, IA, USA. [12]Department of Physics and Astronomy, University of California at Irvine, Irvine, CA, USA. [13]School of Physics, University of Hyderabad, Hyderabad, India. [14]Faculty of Engineering and Natural Sciences, Engineering Sciences Department, Bandırma Onyedi Eylül University, Bandırma, Turkey. [15]School of Physics and Astronomy, University of Minnesota Twin Cities, Minneapolis, MN, USA. [16]Department of Physics, Panjab University, Chandigarh, India. [17]Department of Physics, IIT Guwahati, Guwahati, India. [18]Particle Physics Research Centre, Department of Physics and Astronomy, Queen Mary University of London, London, UK. [19]Department of Physics, IIT Hyderabad, Hyderabad, India. [20]Department of Physics and Astronomy, Michigan State University, East Lansing, MI, USA. [21]Department of Physics, Colorado State University, Fort Collins, CO, USA. [22]Institute for Nuclear Research of the Russian Academy of Sciences, Moscow, Russia. [23]Physics and Astronomy Department, University College London, London, UK. [24]Department of Physics, University of Texas at Austin, Austin, TX, USA. [25]Department of Physics, University of Wisconsin–Madison, Madison, WI, USA. [26]Department of Physics, William & Mary, Williamsburg, VA, USA. [27]Illinois Institute of Technology, Chicago, IL, USA. [28]Department of Physics and Astrophysics, University of Delhi, Delhi, India. [29]Department of Physics, Southern Methodist University, Dallas, TX, USA. [30]Department of Physics, University of Virginia, Charlottesville, VA, USA. [31]Argonne National Laboratory, Argonne, IL, USA. [32]Department of Physics, University of Houston, Houston, TX, USA. [33]Department of Mathematics, Statistics and Physics, Wichita State University, Wichita, KS, USA. [34]Institute of Physics, The Czech Academy of Sciences, Prague, Czech Republic. [35]Department of Physics, University of South Alabama, Mobile, AL, USA. [36]Department of Physics and Astronomy, Tufts University, Medford, MA, USA. [37]Department of Physics and Astronomy, University of Minnesota Duluth, Duluth, MN, USA. [38]Institute of Computer Science, The Czech Academy of Sciences, Prague, Czech Republic. [39]Department of Physics, Syracuse University, Syracuse, NY, USA. [40]Department of Physics and Astronomy, University of South Carolina, Columbia, SC, USA. [41]Czech Technical University in Prague, Prague, Czech Republic. [42]Department of Physics, Cochin University of Science and Technology, Kochi, India. [43]California Institute of Technology, Pasadena, CA, USA. [44]Department of Physics, University of Pittsburgh, Pittsburgh, PA, USA. [45]National Institute of Science Education and Research, Khurda, India. [46]Florida State University, Tallahassee, FL, USA. [47]Department of Physics, Institute of Science, Banaras Hindu University, Varanasi, India. [48]Faculty of Mathematics and Physics, Institute of Particle and Nuclear Physics, Charles University, Prague, Czech Republic. [49]Ohio State University, Columbus, OH, USA.

## The T2K Collaboration

K. Abe[50], S. Abe[50], H. Adhkary[51], R. Akutsu[52], H. Alarakia-Charles[53], Y. I. Alj Hakim[54], S. Alonso Monsalve[55], L. Anthony[56], S. Aoki[57], K. A. Apte[56], T. Arai[58], T. Arihara[59], S. Arimoto[60], Y. Ashida[61], E. T. Atkin[56], N. Babu[62], V. Baranov[5], G. J. Barker[63], G. Barr[64], D. Barrow[64], P. Bates[65], L. Bathe-Peters[64], M. Batkiewicz-Kwasniak[66], N. Baudis[64], V. Berardi[67], L. Berns[61], S. Bhattacharjee[62], A. Blanchet[58], A. Blondel[69,70], P. M. M. Boistier[71], S. Bolognesi[71], S. Bordoni[69], S. B. Boyd[63], C. Bronner[72], A. Bubak[73], M. Buizza Avanzini[74], J. A. Caballero[75], F. Cadoux[69], N. F. Calabria[67], S. Cao[76], S. Cap[69], D. Carabadjac[74,77], S. L. Cartwright[54], M. P. Casado[78,79], M. G. Catanesi[67], J. Chakrani[80], A. Chalumeau[70], D. Cherdack[32], A. Chvirova[22], J. Coleman[65], G. Collazuol[81], F. Cormier[82], A. A. L. Craplet[56], A. Cudd[83], D. D'ago[81], C. Dalmazzone[70], T. Daret[71], P. Dasgupta[84], C. Davis[85], Yu. I. Davydov[5], P. de Perio[86], G. De Rosa[87], T. Dealtry[53], C. Densham[88], A. Dergacheva[22], R. Dharmapal Banerjee[89], F. Di Lodovico[90], G. Diaz Lopez[70], J. Dolan[68], D. Douqa[69], T. A. Doyle[91], O. Drapier[74], K. E. Duffy[64], J. Dumarchez[70], P. Dunne[56], K. Dygnarowicz[51], A. Eguchi[58], J. Elias[93], S. Emery-Schrenk[71], G. Erofeev[22], A. Ershova[71], G. Eurin[71], D. Fedorova[22], S. Fedotov[22], M. Feltre[81], L. Feng[60], D. Ferlewicz[52], A. J. Finch[53], M. D. Fitton[88], M. Forza[81], C. Forza[81], M. Friend[52,94], Y. Fujii[52], Y. Fukuda[95], Y. Furui[59], J. García-Marcos[96], A. C. Germer[85], L. Giannessi[69],

C. Giganti[70], M. Girgus[51], V. Glagolev[5], M. Gonin[97], R. González Jiménez[75], J. González Rosa[75], E. A. G. Goodman[98], K. Gorshanov[22], P. Govindaraj[51], M. Grassi[81], M. Guigue[70], F. Y. Guo[91], D. R. Hadley[63], S. Han[60,99], D. A. Harris[100], R. J. Harris[53,88], T. Hasegawa[52,94], C. M. Hasnip[68], S. Hassani[71], N. C. Hastings[52], Y. Hayato[50,86], I. Heitkamp[61], D. Henaff[71], Y. Hino[52], J. Holeczek[73], A. Holin[88], T. Holvey[64], N. T. Hong Van[101], T. Honjo[102], M. C. F. Hooft[96], K. Hosokawa[50], J. Hu[60], A. K. Ichikawa[61], M. Ieki[50], M. Ikeda[50], T. Ishida[52,94], T. Ishitsuka[103], A. Izmaylov[22], N. Jachowicz[96], S. J. Jenkins[65], C. Jesús-Valls[86], M. Jia[91], J. J. Jiang[91], J. Y. Ji[91], T. P. Jones[53], P. Jonsson[56], S. Joshi[71], C. K. Jung[91], M. Kabirnezhad[56], A. C. Kaboth[100], H. Kakuno[59], J. Kameda[50], S. Karpova[69], V. S. Kasturi[69], Y. Kataoka[50], T. Katori[90], Y. Kawamura[102], M. Kawaue[60], E. Kearns[86,105], M. Khabibullin[22], A. Khotjantsev[22], T. Kikawa[60], S. King[90], V. Kiseeva[5], J. Kisiel[73], A. Klustová[56], L. Kneale[54], H. Kobayashi[58], L. Koch[106], S. Kodama[58], M. Kolupanova[22], A. Konaka[82], L. L. Kormos[53], Y. Koshio[86,107], C. Lin[96], R. P. Litchfield[98], Y. Kudenko[22], Y. Kudo[72], A. Kumar Jha[96], R. Kurjata[92], V. Kurochka[22], T. Kutter[62], L. Labarga[109], M. Lachat[93], K. Lachner[55], J. Lagoda[108], S. M. Lakshmi[73], M. Lamers James[63], A. Langella[87], D. H. Langridge[104], J.-F. Laporte[71], D. Last[93], N. Latham[90], M. Laveder[81], L. Lavitola[87], M. Lewis[53], D. Leon Silverio[110], S. Levorato[102], S. V. Lewis[90], B. Li[55], C. Lin[96], R. P. Litchfield[98], S. L. Liu[91], W. Li[64], A. Longhin[81], A. Lopez Moreno[90], L. Ludovici[111], X. Lu[63], T. Lux[78], L. N. Machado[98], L. Magaletti[67], K. Mahn[20], M. Mandal[108], S. Manly[93], A. D. Marino[83], D. G. R. Martin[56], D. A. Martinez Caicedo[110], L. Martinez[78], M. Martini[70,112], T. Matsubara[52], R. Matsumoto[113], V. Matveev[22], C. Mauger[85], K. Mavrokoridis[65], N. McCauley[65], K. S. McFarland[93], C. McGrew[91], J. McKean[56], A. Mefodiev[22], G. D. Megias[75], L. Mellet[20], C. Metelko[65], M. Mezzetto[81], S. Miki[50], V. Mikola[98], E. W. Miller[78], A. Minamino[72], O. Mineev[22], S. Mine[50,114], J. Mirabito[105], M. Miura[50,86], S. Moriyama[50,86], S. Moriyama[72], P. Morrison[98], Th. A. Mueller[74], D. Munford[32], A. Muñoz[74,97], L. Munteanu[68], Y. Nagai[84], T. Nakadaira[52,94], K. Nakagiri[58], M. Nakahata[50,86], Y. Nakajima[58], B. Nakamura[5], T. Nakamura[60], N. Nakano[115], S. Nakayama[50,86], T. Nakaya[60,86], K. Nakayoshi[52,94], C. E. R. Naseby[56], D. T. Nguyen[116], V. Q. Nguyen[74], K. Niewczas[96], S. Nishimori[52], Y. Nishimura[117], Y. Noguchi[50], T. Nosek[108], F. Nova[88], J. C. Nugent[56], H. M. O'Keeffe[53], L. O'Sullivan[106], R. Okazaki[117], W. Okinaga[58], K. Okumura[86,99], T. Okusawa[102], N. Onda[60], N. Ospina[67], L. Osu[74], Y. Oyama[52,94], V. Paolone[118], J. Pasternak[56], D. Payne[65], M. Pfaff[56], L. Pickering[88], B. Popov[5,70], A. J. Portocarrero Yrey[52], M. Posiadala-Zezula[51], Y. S. Prabhu[51], H. Prasad[89], F. Pupilli[81], B. Quilain[74,97], P. T. Quyen[76,119], E. Radicioni[67], B. Radics[100], M. A. Ramirez[85], R. Ramsden[90], P. N. Ratoff[53], M. Reh[83], G. Reina[98], C. Riccio[91], D. W. Riley[98], E. Rondio[108], S. Roth[120], N. Roy[100], A. Rubbia[55], L. Russo[70], A. Rychter[92], W. Saenz[70], K. Sakashita[52,94], S. Samani[69], F. Sánchez[69], E. M. Sandford[65], Y. Sato[103], T. Schefke[62], C. M. Schloesser[69], K. Scholberg[86,121], M. Scott[56], Y. Seiya[102,122], T. Sekiguchi[52,94], H. Sekiya[50,86], T. Sekiya[59], D. Seppala[20], D. Sgalaberna[55], A. Shaikhiev[22], M. Shiozawa[50,86], Y. Shiraishi[107], A. Shvartsman[22], N. Skrobova[22], K. Skwarczynski[104], D. Smyczek[120], M. Smy[114], J. T. Sobczyk[89], H. Sobel[86,114], F. J. P. Soler[98], A. J. Speers[53], R. Spina[106], A. Srivastava[106], P. Stowell[54], Y. Stroke[22], I. A. Suslov[5], A. Suzuki[57], S. Y. Suzuki[52,94], M. Tada[52,94], S. Tairafune[61], A. Takeda[50], A. Teklu[91], Y. Takeuchi[57,86], H. K. Tanaka[50,86], H. Tanigawa[52], V. V. Tereshchenko[5], N. Thamm[120], C. Touramanis[65], N. Tran[60], T. Tsukamoto[52,94], M. Tzanov[62], Y. Uchida[56], M. Vagins[86,114], M. Varghese[78], I. Vasilyev[22], G. Vasseur[71], E. Villa[68,69], U. Virginet[70], T. Vladisavljevic[88], T. Wachala[66], D. Wakabayashi[61], H. T. Wallace[54], J. G. Walsh[20], L. Wan[105], D. Wark[64,88], M. O. Wascko[64,88], A. Weber[106], R. Wendell[60], M. J. Wilking[123], C. Wilkinson[80], J. R. Wilson[90], K. Wood[80], C. Wret[56], J. Xia[124], K. Yamamoto[102,122], T. Yamamoto[102], C. Yanagisawa[91,125], Y. Yang[64], T. Yano[50], N. Yershov[22], U. Yevarouskaya[91], M. Yokoyama[58,86], Y. Yoshimoto[58], N. Yoshimura[60], R. Zaki[100], A. Zalewska[66], J. Zalipska[108], G. Zarnecki[66], J. Zhang[82,126], X. Y. Zhao[55], H. Zheng[91], H. Zhong[57], T. Zhu[56], M. Ziembicki[92], E. D. Zimmerman[83], M. Zito[70] & S. Zsoldos[90]

[50]University of Tokyo, Institute for Cosmic Ray Research, Kamioka Observatory, Kamioka, Japan. [51]Faculty of Physics, University of Warsaw, Warsaw, Poland. [52]High Energy Accelerator Research Organization (KEK), Tsukuba, Japan. [53]Physics Department, Lancaster University, Lancaster, United Kingdom. [54]School of Mathematical and Physical Sciences, University of Sheffield, Sheffield, United Kingdom. [55]Institute for Particle Physics and Astrophysics, ETH Zurich, Zurich, Switzerland. [56]Department of Physics, Imperial College London, London, United Kingdom. [57]Kobe University, Kobe, Japan. [58]Department of Physics, University of Tokyo, Tokyo, Japan. [59]Department of Physics, Tokyo Metropolitan University, Tokyo, Japan. [60]Department of Physics, Kyoto University, Kyoto, Japan. [61]Faculty of Science, Department of Physics, Tohoku University, Miyagi, Japan. [62]Department of Physics and Astronomy, Louisiana State University, Baton Rouge, LA, USA. [63]Department of Physics, University of Warwick, Coventry, UK. [64]Department of Physics, Oxford University, Oxford, UK. [65]Department of Physics, University of Liverpool, Liverpool, UK. [66]The Henryk Niewodniczanski Institute of Nuclear Physics, Polish Academy of Sciences, Cracow, Poland. [67]Dipartimento Interuniversitario di Fisica, Università e Politecnico di Bari and INFN Sezione di Bari, Bari, Italy. [68]European Organization for Nuclear Research (CERN), Geneva, Switzerland. [69]DPNC, Section de Physique, University of Geneva, Geneva, Switzerland. [70]Laboratoire de Physique Nucléaire et de Hautes Energies (LPNHE), Sorbonne Université, CNRS/IN2P3, Paris, France. [71]IRFU, CEA, Université Paris-Saclay, Gif-sur-Yvette, France. [72]Department of Physics, Yokohama National University, Yokohama, Japan. [73]Institute of Physics, University of Silesia, Katowice, Poland. [74]Laboratoire Leprince-Ringuet, Ecole Polytechnique, IN2P3-CNRS, Palaiseau, France. [75]Departamento de Física Atómica, Molecular y Nuclear, Universidad de Sevilla, Sevilla, Spain. [76]Institute for Interdisciplinary Research in Science and Education (IFIRSE), International Centre for Interdisciplinary Science and Education, Quy Nhon, Vietnam. [77]Université Paris-Saclay, Gif-sur-Yvette, France. [78]Institut de Fisica d'Altes Energies (IFAE), The Barcelona Institute of Science and Technology, Universitat Autònoma de Barcelona, Barcelona, Spain. [79]Departament de Fisica, Universitat Autònoma de Barcelona, Barcelona, Spain. [80]Lawrence Berkeley National Laboratory, Berkeley, CA, USA. [81]Dipartimento di Fisica, INFN Sezione di Padova, Università di Padova, Padova, Italy. [82]TRIUMF, Vancouver, BC, Canada. [83]Department of Physics, University of Colorado Boulder, Boulder, CO, USA. [84]Department of Atomic Physics, Eötvös Loránd University, Budapest, Hungary. [85]Department of Physics and Astronomy, University of Pennsylvania, Philadelphia, PA, USA. [86]Kavli Institute for the Physics and Mathematics of the Universe (WPI), The University of Tokyo Institutes for Advanced Study, University of Tokyo, Kashiwa, Japan. [87]Dipartimento di Fisica, INFN Sezione di Napoli, Università di Napoli, Napoli, Italy. [88]STFC, Rutherford Appleton Laboratory, Didcot, UK. [89]Faculty of Physics and Astronomy, Wroclaw University, Wroclaw, Poland. [90]Department of Physics, King's College London, London, UK. [91]Department of Physics and Astronomy, State University of New York at Stony Brook, Stony Brook, NY, USA. [92]Institute

of Radioelectronics and Multimedia Technology, Warsaw University of Technology, Warsaw, Poland. [93]Department of Physics and Astronomy, University of Rochester, Rochester, NY, USA. [94]Japan Proton Accelerator Research Complex, Tokai, Japan. [95]Department of Physics, Miyagi University of Education, Sendai, Japan. [96]Department of Physics and Astronomy, Ghent University, Gent, Belgium. [97]ILANCE, CNRS, University of Tokyo International Research Laboratory, Kashiwa, Japan. [98]School of Physics and Astronomy, University of Glasgow, Glasgow, UK. [99]Research Center for Cosmic Neutrinos, Institute for Cosmic Ray Research, University of Tokyo, Kashiwa, Japan. [100]Department of Physics and Astronomy, York University, Toronto, Ontario, Canada. [101]International Centre of Physics, Institute of Physics (IOP), Vietnam Academy of Science and Technology (VAST), Hanoi, Vietnam. [102]Department of Physics, Osaka Metropolitan University, Osaka, Japan. [103]Department of Physics, Tokyo University of Science, Faculty of Science and Technology, Noda, Chiba, Japan. [104]Department of Physics, Royal Holloway University of London, Egham, UK. [105]Department of Physics, Boston University, Boston, MA, USA. [106]Institut für Physik, Johannes Gutenberg-Universität Mainz, Mainz, Germany. [107]Department of Physics, Okayama University, Okayama, Japan. [108]National Centre for Nuclear Research, Warsaw, Poland. [109]Department of Theoretical Physics, University Autonoma Madrid, Madrid, Spain. [110]South Dakota School of Mines and Technology, Rapid City, SD, USA. [111]INFN Sezione di Roma and Università di Roma "La Sapienza", Roma, Italy. [112]IPSA-DRII, Ivry-sur-Seine, France. [113]Department of Physics, Institute of Science Tokyo, Tokyo, Japan. [114]Department of Physics and Astronomy, University of California, Irvine, Irvine, CA, USA. [115]Department of Physics, University of Toyama, Toyama, Japan. [116]VNU University of Science, Vietnam National University, Hanoi, Vietnam. [117]Department of Physics, Keio University, Kanagawa, Japan. [118]Department of Physics and Astronomy, University of Pittsburgh, Pittsburgh, PA, USA. [119]Graduate University of Science and Technology, Vietnam Academy of Science and Technology, Hanoi, Vietnam. [120]III. Physikalisches Institut, RWTH Aachen University, Aachen, Germany. [121]Department of Physics, Duke University, Durham, NC, USA. [122]Nambu Yoichiro Institute of Theoretical and Experimental Physics, Osaka, Japan. [123]School of Physics and Astronomy, University of Minnesota, Minneapolis, MN, USA. [124]SLAC National Accelerator Laboratory, Stanford University, Menlo Park, CA, USA. [125]Science Department, Borough of Manhattan Community College, City Univerisity of New York, New York, NY, USA. [126]Department of Physics and Astronomy, University of British Columbia, Vancouver, British Columbia, Canada.

## Methods

### The NOvA experiment

The NOvA experiment measures neutrino oscillations using two detectors of functionally identical construction located along the NuMI neutrino beam[50] produced at the Fermi National Accelerator Laboratory (Fermilab).

The smaller 0.3-kt near detector is located on the Fermilab campus 1 km downstream from the neutrino production target, whereas the 14-kt far detector is located 810 km away in northern Minnesota. The detectors themselves are highly segmented tracking calorimeters consisting of long PVC cells filled with a mineral-oil-based liquid scintillator. Each cell measures 6.6 cm × 3.9 cm in cross-section, runs the full height or width of the detector (15.5 m for the far detector and 3.9 m for the near detector) and is instrumented with a wavelength-shifting fibre and avalanche photodiode to detect the scintillation light produced when charged particles pass through the cell. The cells are arranged in a series of layers, each with either horizontal or vertical orientation, with the direction alternating between layers to provide three-dimensional (3D) event reconstruction. This segmented design offers the excellent muon and electron classification needed for tagging the incoming neutrino flavour. In particular, electromagnetic showers at typical NOvA energies are much larger than the detector cell widths and thus are well-imaged and distinct from many potential backgrounds. The detectors of NOvA are centred 14.6 mrad off the central axis of the NuMI beam, yielding a narrow-band neutrino beam peaked at 1.8 GeV.

As is typical for particle physics experiments, NOvA makes use of detailed simulations of beam production, neutrino interaction physics and detector response as part of the analysis. Given the matching near and far detectors, NOvA forms its oscillation-dependent predictions of the far-detector event rates directly from data using the millions of neutrino interactions recorded in the near detector. This near-to-far extrapolation process is carried out as a function of multiple kinematic and event classification variables. Uncertainties from the simulations have substantially reduced impact as they enter the oscillation fit only to the extent that they affect the mapping between expected near and far event rates, not the event rates of the individual detectors themselves. Uncertainties on the simulations are taken as the a priori uncertainties from, for instance, the external model constraints or other external data and are supplemented by additional model uncertainties in which a priori coverage was deemed unsatisfactory.

Far-detector data are fitted to the corresponding far-detector predictions to extract oscillation parameter constraints. These data are separated by beam mode (that is, neutrino- or antineutrino-dominated running) and further into $\nu_\mu/\bar{\nu}_\mu$ charged current and $\nu_e/\bar{\nu}_e$ charged current candidate samples using a convolutional neural network[51] whose inputs are the calibrated event images recorded by the detector cells. Subsequent reconstruction of tracks and showers within each event provides kinematic information such as estimated neutrino energy. Far detector $\nu_\mu/\bar{\nu}_\mu$ samples are analysed in bins of neutrino energy and hadronic energy fraction. The $\nu_e/\bar{\nu}_e$ samples are analysed in bins related to event containment, event classification score and neutrino energy. More details on the analysis techniques, simulation packages, systematic uncertainties and the overall NOvA experimental design can be found in ref. 13 and the references therein.

### The T2K experiment

The T2K experiment is composed of the J-PARC neutrino beam, a near site with multiple detectors and the water Cherenkov detector Super-Kamiokande (SK) as the far detector. Full details of the experiment can be found in ref. 25.

The primary detector at the near site, 280 m from the target, is a magnetized off-axis (centred at 43.6 mrad) tracking detector called ND280. While taking the data used in this analysis, ND280 consisted of a $\pi^0$ detector followed by a tracker consisting of three time-projection chambers interleaved with two hydrocarbon fine-grained detectors (FGD1 and FGD2), all surrounded by an electromagnetic calorimeter. The stability and direction of the neutrino beam are monitored using the on-axis near detector INGRID.

SK is situated 295 km downstream of the neutrino production target, 43.6 mrad off-axis, and contains 50 kt of water. An inner detector (ID) using 11,129 inward-facing 20-inch photomultiplier tubes (PMTs) detects Cherenkov radiation from charged particles traversing the detector. An optically separated outer detector uses 1,885 outward-facing 8-inch PMTs to reject interactions originating outside the ID volume. SK can discriminate between electrons and muons by their Cherenkov ring profiles.

T2K uses a forward-fitting analysis strategy. First, a model that predicts the event spectra at the near and far detectors is defined and tuned to external experimental data. The predictions are generated by simulating the neutrino flux and cross-section as well as the detector response. The model, with variable parameters, is fit to the ND280 data to obtain tuned values of the parameters with uncertainties. The constrained model resulting from this near-detector fit is then used to make SK predictions, which are fit to the SK data to extract oscillation parameters. Complete details for this analysis, including model details, are in ref. 14.

T2K splits data at the near and far detectors into mutually exclusive samples defined by particle identification in each beam mode. At ND280, events are categorized into 18 samples, nine samples in each of FGD1 and FGD2. In neutrino mode, data with one negatively charged muon is split into three samples in each FGD corresponding to the number of pions (0, 1, or >1). In antineutrino mode, data are first split by whether a negatively or positively charged muon is present, and then divided by the number of pions as in the neutrino-mode data, forming six samples in each FGD. For all samples, the data are fit in a 2D space of the muon momentum and the angle between the muon and the average beam direction. The exclusive samples allow the near-detector fit to better constrain parameters related to different neutrino–nucleus interaction modes. At SK, the data are divided into three samples in neutrino mode: one-ring muon-like, one-ring electron-like and one-ring electron-like with one decay electron; in antineutrino mode, only the one-ring muon-like and one-ring electron-like samples are used. The data are binned in reconstructed neutrino energy. All electron-like samples are additionally binned in a second dimension, the angle between the reconstructed electron direction and the beam direction.

Detector systematic uncertainties are evaluated using a variety of sideband samples and calibrations, covering effects such as particle identification, particle momentum reconstruction, secondary particle interactions and fiducial volume effects.

### Correlations in flux modelling

The modelling of the neutrino flux depends on many details relating to the incident proton beam, the hadron production target and the magnetic focusing horns. As these details are specific to each experiment, flux systematic uncertainties due to magnetic field variations, component alignment and other beamline properties are uncorrelated between the experiments.

The only possible correlation identified was the pion and kaon production models and the use of hadron interaction experiments to tune them[52,53]. In the case of NOvA, the primary data are from the NA49 experiment[33], which collected thin-target (slices of the target material) data at 158 GeV $c^{-1}$, which is then scaled to the NuMI beam energy. The NA61/SHINE experiment, which collected data for T2K, uses some of the same detectors and the same beamline as NA49. NA61/SHINE[34,35] collected both thin-target and replica-target (a full-sized target) data for T2K at 31 GeV $c^{-1}$, the J-PARC beam momentum. Checking the consistency of the NA49 and NA61/SHINE data used is difficult, as the data are collected at different beam energies.

The NOvA experiment primarily uses thin-target NA49 hadron production data to tune the particle multiplicities, reweighting interactions and particle propagation inside the target and other beamline materials. By contrast, T2K uses thin and replica-target data from NA61/SHINE to reweight the multiplicities of particles exiting the target. Given these differences in data collection and tuning methodology, and given that flux uncertainties have a suppressed influence after ND data constraints are considered, there is no expectation of significant correlations between flux systematic parameters for NOvA and T2K in the joint fit.

## Correlations in detector modelling

The experiments use different detector technologies as well as strategies for forming data samples, which removes most opportunities for correlation. However, the modelling of particle propagation through the detectors derives from the same underlying physics. This propagation is called secondary interaction (SI), and the case of pion SI is noteworthy, as this process is expected to occur in both experiments, and for T2K, it is an important effect. T2K selects exclusive data samples in which a change in reconstructed pion multiplicity can cause migration between samples. By contrast, NOvA uses inclusive selections, and pion SI has minimal effect on the calorimetric energy estimation at NOvA. Thus, we do not expect significant correlations due to pion SI.

## Tests of individual parameter correlations

Neutrino-on-nucleus scattering plays a central part in both experiments, but the modelling of this physics has substantial differences between the two individual analyses. These differences, together with the presence of different nuclear targets, neutrino energies and near-detector strategies, mean that direct estimation of systematic uncertainty correlations in the neutrino scattering models is highly non-trivial. As part of this analysis, we tested how significant inter-experimental systematic uncertainty correlations could be, starting by identifying the most impactful systematic uncertainties of T2K and NOvA and exploring correlations between them.

To determine an impactful systematic parameter, we carry out a fit to pseudo-data generated with all parameters at their prior values from our nominal model. Then, for each parameter in turn, we reweight all steps from the obtained MCMC chain to have a tight ('shrunk') prior for that parameter around a different value ('pulled') to that used to generate the pseudo-data and study the change in the extracted oscillation parameter intervals. This procedure mocks up the result of an external experiment, providing a strong constraint on each systematic parameter at a different value from that preferred by simulated pseudo-data. This 'shrink and pull' study allows for assessing the single-parameter impact on the systematic uncertainty and the estimated credible intervals of the measurement of the individual neutrino oscillation parameters.

First, we identify both the systematic parameters of NOvA and T2K with the largest impact on $\delta_{CP}$, $\sin^2\theta_{23}$ and $\Delta m_{32}^2$ in the joint fit.

For both experiments, the largest change in $\delta_{CP}$ credible interval comes from uncertainties on $\nu_e$ and $\bar{\nu}_e$ normalizations. As discussed, these uncertainties are implemented identically in both experiments, and we have correlated them in the joint analysis. No additional interaction uncertainties in our models have any significant impact on the resulting credible intervals of $\delta_{CP}$.

For $\sin^2\theta_{23}$, all the individual interaction systematic parameters have very small effects, changing the width of the $1\sigma$ interval by less than 2% when shrunk by 50% and pulled $1\sigma$ away from the nominal value. The largest change in credible interval comes from the uncertainty on the neutron visible energy for NOvA, and the two-particle two-hole (2p2h) C/O cross-section scale for T2K (2p2h C/O cross-section scale allows the 2p2h cross-section on carbon to differ from that for oxygen). For $\Delta m_{32}^2$, all the individual interaction parameters have a negligible effect on the resulting $\Delta m_{32}^2$ credible intervals. Hence, we widened the list of

considered parameters and identified the calorimetric energy scale uncertainty of NOvA and the SK energy scale uncertainty of T2K as the most impactful for $\Delta m_{32}^2$.

Second, despite there being no a priori reason to expect correlations between these specific parameters, we test whether or not correlating the most impactful T2K parameter with the most impactful NOvA parameter modifies oscillation parameter constraints in the joint fit in a significant way. We simulate pseudo-data to which we perform a joint fit while treating the T2K and NOvA parameters described above as either uncorrelated, fully correlated or fully anticorrelated. We repeat the study for each pair of the most impactful parameters of T2K and NOvA with respect to $\delta_{CP}$, $\sin^2\theta_{23}$ and $\Delta m_{32}^2$. In the case of $\Delta m_{32}^2$, we further inflate the original SK energy scale uncertainty from 2% to 7% to amplify the effect. Finally, we check the extracted $1\sigma$ and $2\sigma$ credible regions for any substantial differences between the three correlation configurations. These tests are repeated for three sets of pseudo-data generated with oscillation parameter values that are T2K-like, NOvA-like and NuFit-like[54], which are chosen to be close to recent data results from the respective collaborations and are given in Extended Data Table 1.

As an example, Extended Data Fig. 1 shows the results in terms of the posterior probability distributions and credible regions of the parameters of interest from the set of fits with the largest single-parameter impact on $\sin^2\theta_{23}$. We conclude that the choice of correlation between single parameters does not significantly change the oscillation parameter constraints derived from the current version of the joint analysis.

## Nightmare parameters

As described in the main text, we study correlations in more extreme situations using the so-called nightmare parameters, which are either artificially constructed parameters or existing parameters with highly inflated uncertainties chosen to be deliberately problematic for the individual analyses. The prior uncertainties of the parameters are set so that they are comparable in impact to the statistical uncertainties on the measurements under study. We carry out this procedure separately for simulated measurements of $\Delta m_{32}^2$ and $\theta_{23}$. No nightmare study was carried out for $\delta_{CP}$ because its total systematic uncertainty compared with the statistical uncertainty is much smaller than for the other two cases.

We construct pseudo-datasets with both the NOvA and T2K nightmare parameters shifted by one standard deviation from their prior values, inducing a systematic bias representing a simultaneous and coordinated shift in both NOvA and T2K data. We fit this pseudo-data while treating the NOvA and T2K nightmare parameters as either fully correlated, uncorrelated or anticorrelated. The results of the nightmare parameters correlation study are presented as $1\sigma$ credible 2D regions of $\Delta m_{32}^2$ – $\sin^2\theta_{23}$ in Extended Data Fig. 2 for both nightmare scenarios. We conclude that there is no significant difference in treating the nightmare parameters as either fully correlated (matching the pseudo-data) or uncorrelated between the experiments, whereas the incorrect anticorrelated case yields a clear bias. We note that these are not general conclusions but are specific to the T2K and NOvA analysis versions and cumulative beam exposures used here. The construction of the nightmare parameters is also not a unique choice, and other formulations of the parameters could be considered.

## Out-of-model variations

As described in the main text, we use a set of discrete changes to the base cross-section model to test the robustness of our analysis. For each test, pseudo-data are generated assuming the specific model variation, and these pseudo-data are then fit either with the default analysis directly, which does not incorporate the model variation ('out-of-model' case) or with a modified analysis that has had its nominal event spectra altered to match the spectra expected under the varied model ('in-model' case). Between these two cases, we require that the width of each of the extracted oscillation parameter intervals changes by no more than

10% (representing a small 'error on the error') and that the centre of the interval does not move by more than 50% of the systematic uncertainty (indicating adequate systematic uncertainty coverage of the tested out-of-model variation). Furthermore, we require that taking the largest changes seen across these studies does not affect the stated conclusions on CP violation or mass ordering determination for the analysis.

Three variations were chosen to perform the out-of-model studies:

- MINERvA 1π: this model suppresses charged current (CC) and neutral current (NC) resonant pion production at low $Q^2$ to ensure good agreement between the MINERvA data[55] and the implementation of the Rein–Seghal model in the GENIE v.2 neutrino interaction simulation software[37].
- Non quasi-elastic (non-QE): in the T2K oscillation analysis[14], the ND280 data samples with a muon candidate and zero pion candidates are underpredicted by the pre-fit T2K nominal model by 10% in both FGDs, which the fit accounts for by enhancing the charged current quasi-elastic (CCQE) interaction rate. To check this large freedom does not cause bias, an alternate model is produced, in which this underprediction is attributed to only non-QE processes.
- Pion SI: the pion SI model in the GEANT4 detector simulation toolkit [56] was replaced with the Salcedo–Oset model[57] implemented in the NEUT generator[36], tuned to π–A scattering data[58].

We also used this process to study what happens when fitting pseudo-data constructed for both experiments using the nominal cross-section model of one or the other experiment (T2K-like and NOvA-like studies).

We show example results here for the MINERvA 1π case. Extended Data Fig. 3a,b shows the effect of this alternative model on event spectra used in the analysis. Note that not all event spectra are uniformly binned. Extended Data Figs. 3c–g and 4 compare the in-model and out-of-model fit results. No failures of our criteria are seen in any of the cases. More generally, no significant bias is seen in this joint fit for any of the model variations studied across any of the three tested sets of oscillation parameter values.

Some more recent T2K analyses[45] did see criteria failures when considering an alternative nuclear model, HF-CRPA[59], and as a result widened their $\Delta m_{32}^2$ intervals. Both NOvA and T2K have independently studied the impact of the HF-CRPA model on the analyses used in this joint result, and we estimate that any potential effects in the context of this joint fit are within the thresholds set for our out-of-model variation tests.

## Goodness of fit

The posterior-predictive $P$-value[41] technique is used to determine whether a model provides a good fit to the data it is confronted with. We require that the posterior-predictive $P$-value to obtain the far-detector data in all samples, given the joint post-fit model, is greater than 0.05. We also check the $P$-values for individual far-detector samples and require that they are greater than 0.05 after allowing for the look-elsewhere effect, using the Bonferroni correction[60]. All the $P$-values from the joint fit are shown in Extended Data Table 2. All the $P$-values (both total and split sample by sample) are within our acceptable range (>0.05), even without taking the look-elsewhere effect into account. This means that the model used in this joint fit—that is, the systematic models of the individual experiments with a shared oscillation parameter model—fits our data well, even when looking at individual samples. The $P$-values are consistent with previous T2K-only and NOvA-only analyses. The $P$-value considering rate and shape for all T2K samples in a T2K-only fit is 0.73, whereas the $P$-value considering all T2K samples in the joint fit is 0.75. Similarly, the $P$-values for all NOvA samples are 0.56 (NOvA-only fit) and 0.64 (joint fit).

Example posterior predictions[61] of the spectra for the $\nu_\mu$ and $\nu_e$ sub-samples of both experiments, overlaid over the observed data, are shown in Extended Data Fig. 5.

## Priors

The default priors on the oscillation parameters for this analysis are as follows: flat between $-\pi$ and $\pi$ in $\delta_{CP}$, flat between 0 and 1 in $\sin^2\theta_{23}$, flat in $\Delta m_{32}^2$ and Gaussian with $\mu \pm \sigma = (2.18 \pm 0.07) \times 10^{-2}$ in $\sin^2\theta_{13}$. Where alternate priors are used, this is stated in the text.

This analysis is not sensitive to the oscillation parameters $\sin^2\theta_{12}$ and $\Delta m_{21}^2$ beyond existing experimental constraints; their Gaussian priors are set to be $\sin^2\theta_{12} = 0.307 \pm 0.013$ and $\Delta m_{21}^2 = (7.53 \pm 0.18) \times 10^{-5}$ eV$^2$. These values, along with a Gaussian prior on $\sin^2\theta_{13}$, when it is used, come from the 2020 version of the Particle Data Group (PDG) summary tables[40], which were current at the time of the original analyses. Updates to these constraints in more recent versions of the PDG do not change any conclusions.

As well as the standard prior flat in $\delta_{CP}$, we also studied the effect of a prior flat in $\sin\delta_{CP}$ and saw no significant changes in conclusions.

Moreover, the experiments define priors for all of the systematic parameters in their models. These definitions are detailed in the individual experiment analyses underlying this work.

## Highest posterior probability values and 1σ credible intervals

Extended Data Table 3 summarizes the highest posterior probability values and credible intervals measured jointly by NOvA and T2K.

## Additional oscillation parameter plots

The main text shows the 1D posterior distributions and credible intervals for the Jarlskog invariant, $\delta_{CP}$ and $\sin^2\theta_{23}$, as well as 2D distributions and credible regions for the latter two. In this section, we present the 1D distributions and credible intervals for $\delta_{CP}$, $\sin^2\theta_{23}$, $\sin^2 2\theta_{13}$ and $|\Delta m_{32}^2|$, and 2D distributions and credible regions for all pairwise combinations of these parameters. These are shown in Extended Data Figs. 6–8, for the cases of marginalized over both mass orderings, conditional on the normal ordering and conditional on the inverted ordering, respectively. The distributions and intervals are shown in a triangle plot, in which a lower triangular matrix of plots shows the 1D distributions along the diagonal and the 2D distributions in each of the off-diagonal positions.

## Reactor $\Delta m_{32}^2$

The energy-dependent $\bar\nu_e \to \bar\nu_e$ oscillation probability measured by reactor experiments is sensitive to $|\Delta m_{32}^2|$, and reactor measurements of this parameter are expected to agree with long-baseline measurements only under the correct mass ordering assumption. Under the incorrect ordering assumption, these two techniques are expected to measure incorrect values that differ from one another by about 2–3% (ref. 62). Thus, comparing $|\Delta m_{32}^2|$ measurements from accelerator and reactor experiments under both mass ordering hypotheses can inform mass ordering discrimination. The Daya Bay experiment[47] provides the tightest constraints on $\theta_{13}$ and also reports a 2D $\theta_{13} - \Delta m_{32}^2$ likelihood that we can directly incorporate into our joint fit instead of the $\theta_{13}$-only prior discussed elsewhere in this study.

The mass ordering Bayes factor obtained when using this 2D reactor constraint is 1.4 in favour of the normal ordering, in contrast to 1.3 in favour of the inverted ordering when using the $\theta_{13}$-only reactor constraint. This slight pull towards a preference for the normal ordering is expected, given the relative agreement of the Daya Bay and NOvA+T2K $|\Delta m_{32}^2|$ measurements shown in Fig. 2 (inverted ordering) and Extended Data Fig. 9a (normal ordering). However, there remains no statistically significant mass ordering preference in this combination.

## Additional global comparisons

In Extended Data Fig. 9, results of the analysis using the default priors are compared with other experimental measurements. The statement on $\Delta m_{32}^2$ precision is still valid for the normal ordering assumption. As in the case of the $\sin^2 2\theta_{13}$ result (Extended Data Fig. 9b,c), the

long-baseline measurements (in this comparison, without applying the prior from reactor measurements) are consistent with reactor experiments, with larger consistency in the normal ordering than the inverted ordering. We do not strongly prefer either octant of $\sin^2\theta_{23}$ (Extended Data Fig. 9d,e), which is consistent with other modern experiments. The joint analysis result for $\delta_{CP}$ (Extended Data Fig. 9f,g) is consistent with all experiments and their combinations, although the uncertainty remains large.

## Data availability

Inquiries regarding the data and posteriors used in this result may be directed to the collaborations.

## Code availability

The NOvA and T2K collaborations develop and maintain the code used for the simulation of the experimental apparatus and statistical analysis of the raw data used in this result. This code is shared among the collaborations, but because of the size and complexity of the codebases, it is not publicly distributed. Inquiries regarding the algorithms and methods used in this result may be directed to the collaborations.

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

**Acknowledgements** We thank the Fermi National Accelerator Laboratory (Fermilab), a US Department of Energy, Office of Science, HEP User Facility. Fermilab is managed by the Fermi Forward Discovery Group, acting under contract no. 89243024CSC000002. This work was supported by the US Department of Energy, including through the US–Japan Science and Technology Cooperation Program in HEP; the US National Science Foundation; the Department of Science and Technology, India; the European Research Council; the MSMT CR, GA UK, Czech Republic; the RAS, the Ministry of Science and Higher Education, and RFBR, Russia; CNPq and FAPEG, Brazil; UKRI, STFC and the Royal Society, UK; and the state and University of Minnesota. We are grateful for the contributions of the staff of the University of Minnesota at the Ash River Laboratory and of Fermilab. We thank the J-PARC staff for superb accelerator performance. We thank the CERN NA61/SHINE Collaboration for providing valuable particle production data. We acknowledge the support of MEXT, JSPS KAKENHI and bilateral programmes, Japan; NSERC, the NRC, and CFI, Canada; the CEA and CNRS/IN2P3, France; the Deutsche Forschungsgemeinschaft (DFG 397763730 and 517206441), Germany; the NKFIH (NKFIH 137812 and TKP2021-NKTA-64), Hungary; the INFN, Italy; the Ministry of Science and Higher Education (2023/WK/04) and the National Science Centre (UMO-2018/30/E/ST2/00441, UMO-2022/46/E/ST2/00336 and UMO-2021/43/D/ST2/01504), Poland; the RSF (RSF 24-12-00271) and the Ministry of Science and Higher Education, Russia; MICINN (PID2022-136297NB-I00 /AEI/10.13039/501100011033/ FEDER, UE, PID2021-124050NB-C31, PID2019-104676GB-C33 PID2024-157541NB-I00 (UAM) and PID2023-146401NB-I00 (US), Severo Ochoa Centres of Excellence Programme 2025-2029 (CEX2024001442-S), Government of Andalucia (FQM160) and the University of Tokyo ICRR's Inter-University Research Program FY2025 Ref. J1, and ERDF and European Union (UAM: H2020-MSCA-RISE-GA872549- SK2HK) and NextGenerationEU funds (PRTR-C17.I1) and Generalitat de Catalunya (AGAUR 2021-SGR-01506, CERCA programme) University of Seville grant (RYC2022-035203-I funded by MICIU/AEI/10.13039/501100011033, "ERDF a way of making Europe" and FSE+, Ayudas "Atracción de Investigadores con Alto Potencial". VII Plan Propio de Investigación y Transferencia, 2025, Ref. VIIPPIT-2023-V.4, and Secretariat for Universities and Research of the Ministry of Business and Knowledge of the Government of Catalonia and the European Social Fund (2022FI_B 00336), Spain; the SNSF and SERI, Switzerland; the STFC and UKRI, the UK; the DOE, the USA, including through the US–Japan Science and Technology Cooperation Program in HEP; and NAFOSTED (103.99-2023.144,IZVSZ2.203433), Vietnam. We also thank CERN for the UA1/NOMAD magnet, DESY for the HERA-B magnet mover system, the BC DRI Group, Prairie DRI Group, ACENET, SciNet, and CalculQuebec consortia in the Digital Research Alliance of Canada, and GridPP in the UK, the CNRS/IN2P3 Computing Center in France and NERSC, the USA. Moreover, the participation of individual researchers and institutions has been further supported by funds from the ERC (FP7), 'la Caixa' Foundation, the Horizon 2020 Research and Innovation Programme of the European Union under the Marie Skłodowska–Curie grant; the JSPS, Japan; the Royal Society, the UK; the French ANR and Sorbonne Université Emergences programmes; the VAST-JSPS (no. QTJP01.02/20-22); and the DOE Early Career Program, the USA. For open access, we have applied for a Creative Commons Attribution (CC BY) license to any Author Accepted Manuscript version arising.

**Author contributions** The operation, Monte Carlo simulation and data analysis of the T2K and NOvA experiments are carried out by the T2K and NOvA Collaborations with contributions from all collaborators listed as authors on this paper. The scientific results presented here have been presented to and discussed by the full collaborations, and all authors have approved the final version of the paper.

**Competing interests** The authors declare no competing interests.

**Additional information**
**Correspondence and requests for materials** should be addressed to the NOvA Collaboration or the T2K Collaboration.

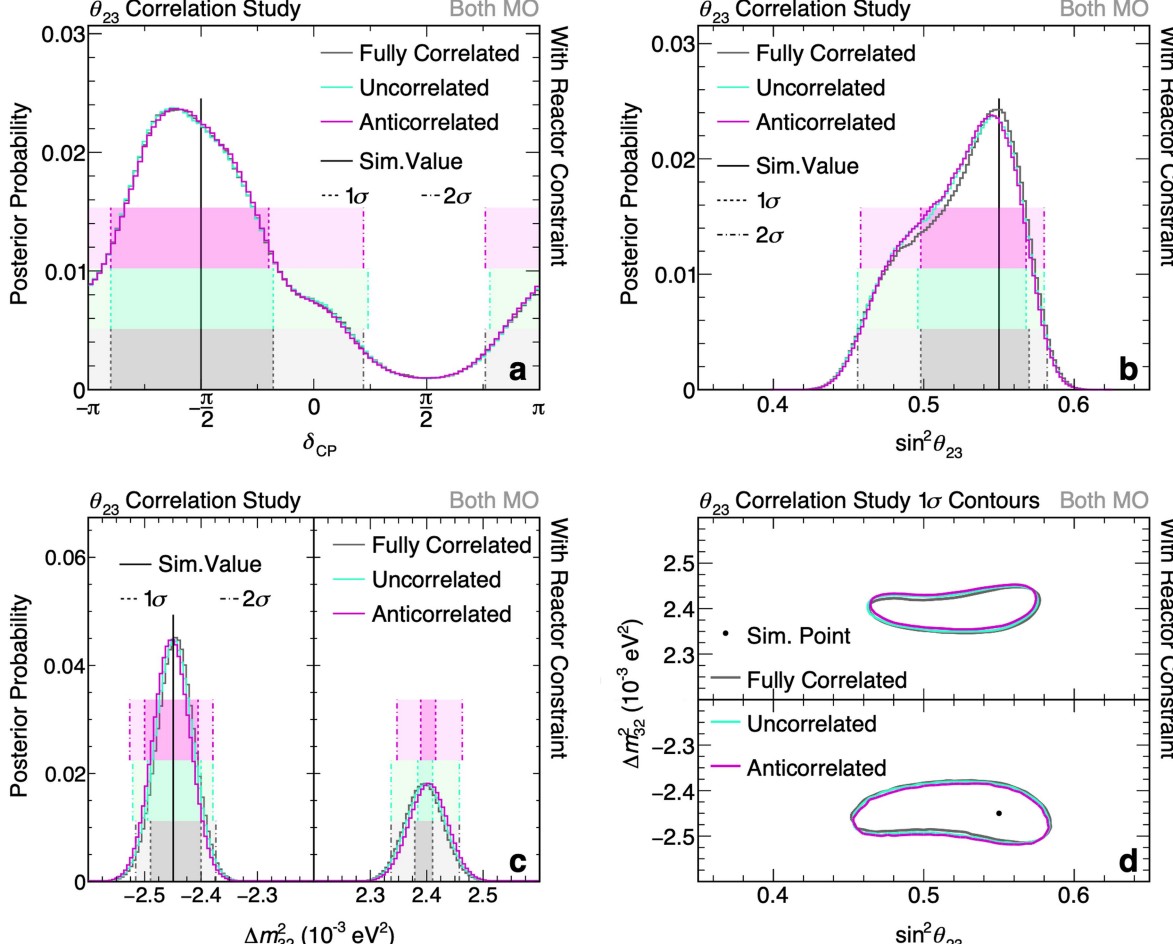

**Extended Data Fig. 1 | Correlation study comparison plots.** Posterior probability distributions of $\delta_{CP}$ (a), $\sin^2\theta_{23}$ (b), and $\Delta m^2_{32}$ (c) and $1\sigma$ credible regions in $\Delta m^2_{32} - \sin^2\theta_{23}$ (d), marginalized over both neutrino mass ordering hypotheses ('Both MO') from fits to pseudo-data simulated with the NuFit-like oscillation parameter values. The fits were run in three configurations while treating the systematic uncertainties with the largest impact on $\sin^2\theta_{23}$ (visible neutron energy and 2p2h C/O scale) as either 100% correlated (gray), uncorrelated (teal), or 100% anticorrelated (magenta). Overlaid with the corresponding $1\sigma$ (dark shaded areas, dashed) and $2\sigma$ (light shaded areas, dash-dotted) credible intervals.

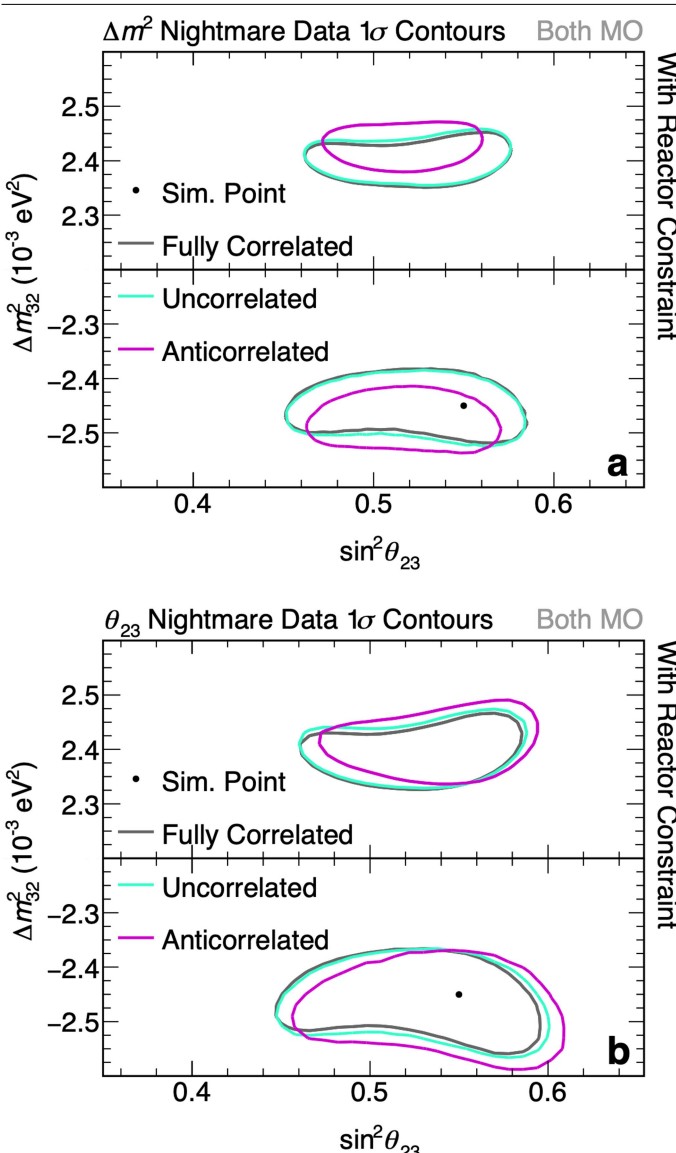

**Extended Data Fig. 2 | 'Nightmare' study comparisons.** $1\sigma$ credible regions in $\Delta m_{32}^2 - \sin^2\theta_{23}$ posterior probability distributions marginalized over both neutrino mass ordering hypotheses ('Both MO') from fits to pseudo-data simulated with the NuFit-like oscillation parameter values and a fully symmetric systematic bias to affect (a) $\Delta m_{32}^2$ ('$\Delta m^2$ nightmare') and (b) $\sin^2\theta_{23}$ ('$\theta_{23}$ nightmare'). The fits were run while treating the NOvA and T2K nightmare parameters as either 100% correlated (gray), uncorrelated (teal), or 100% anticorrelated (magenta).

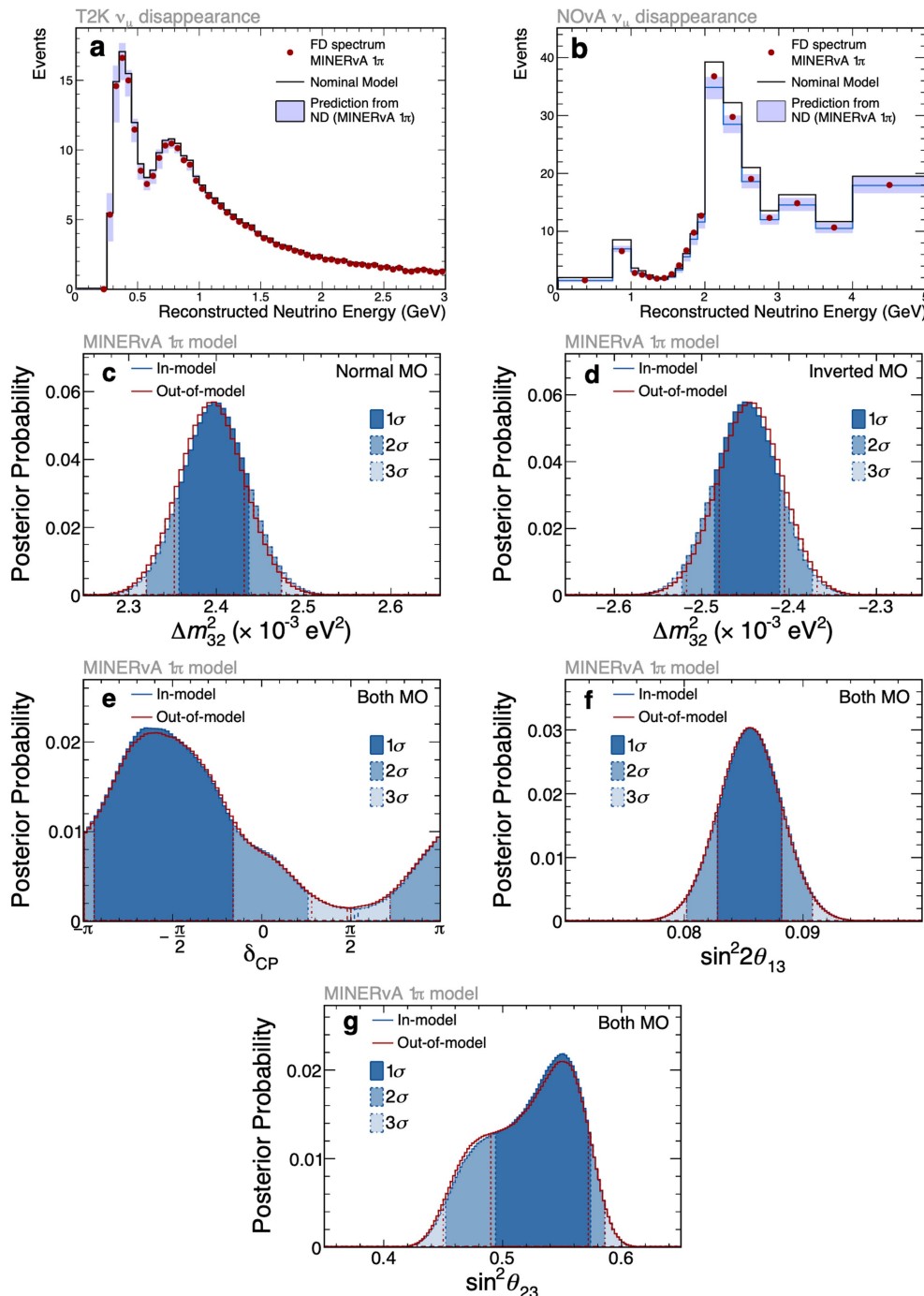

**Extended Data Fig. 3 | Out-of-model study spectra and comparison plots in 1D.** NOvA+T2K out-of-model study with suppressed pion production at low $Q^2$ ('MINERvA 1π' case). The change on the FD pseudo-data and prediction with systematic uncertainties after incorporating the alternate data at the ND is shown for T2K (a) and NOvA (b). Central value of the nominal model is shown for comparison. 1D posterior probability distributions from a fit to pseudo-data generated at the NuFit-like oscillation parameter values are shown for $\Delta m^2_{32}$ marginalized separately over the normal (c) and inverted (d) mass orderings, and for $\delta_{CP}$ (e), $\sin^2 2\theta_{13}$ (f), and $\sin^2 \theta_{23}$ (g) marginalized over both mass orderings. The in-model (blue shaded) and out-of-model (red curve) scenarios are displayed.

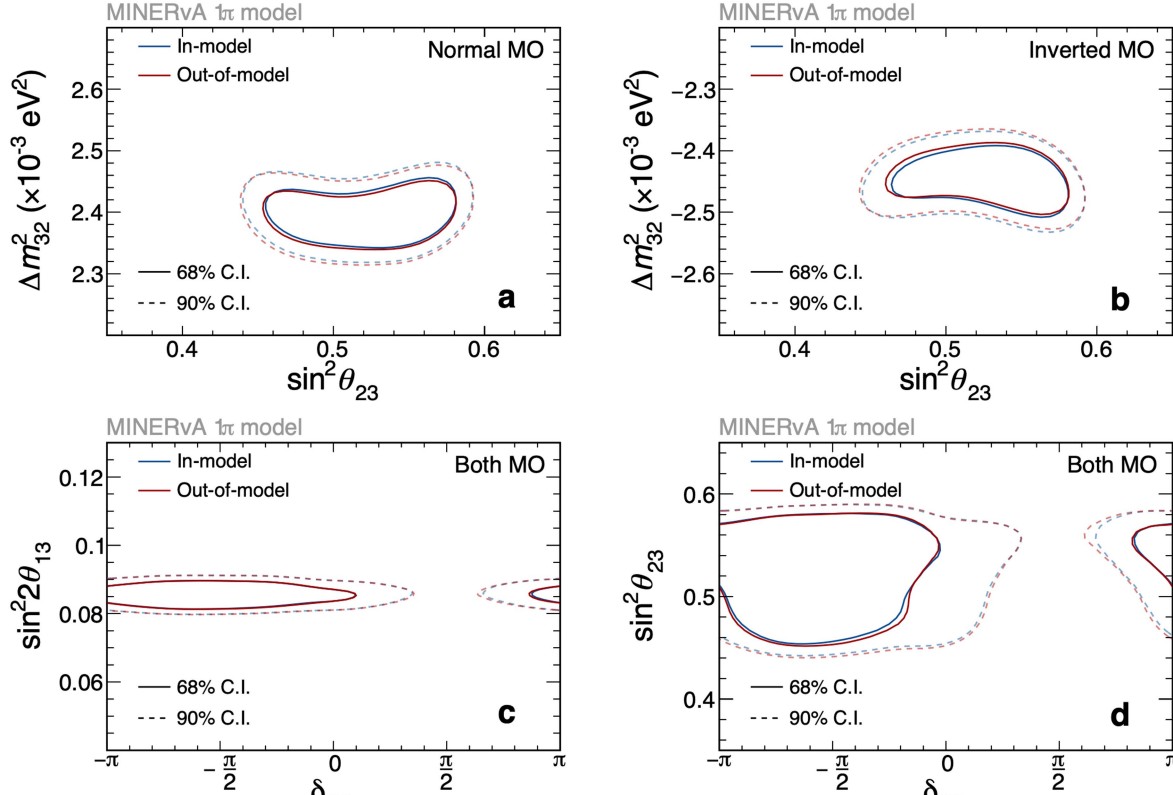

**Extended Data Fig. 4 | Out-of-model study comparison plots in 2D.**
NOvA+T2K out-of-model study with suppressed pion production at low $Q^2$
('MINERvA 1π' case). 68% and 90% contours are shown on the $\sin^2\theta_{23} - \Delta m^2_{32}$
surface marginalized separately over the normal (a) and inverted (b) mass
orderings, and on the surfaces of $\delta_{CP} - \sin^2 2\theta_{13}$ (c) and $\delta_{CP} - \sin^2\theta_{23}$ (d) parameters,
marginalized over both mass orderings, from a fit to pseudo-data generated at
the NuFit-like oscillation parameter values. The in-model (blue shaded) and
out-of-model (red curve) scenarios are shown.

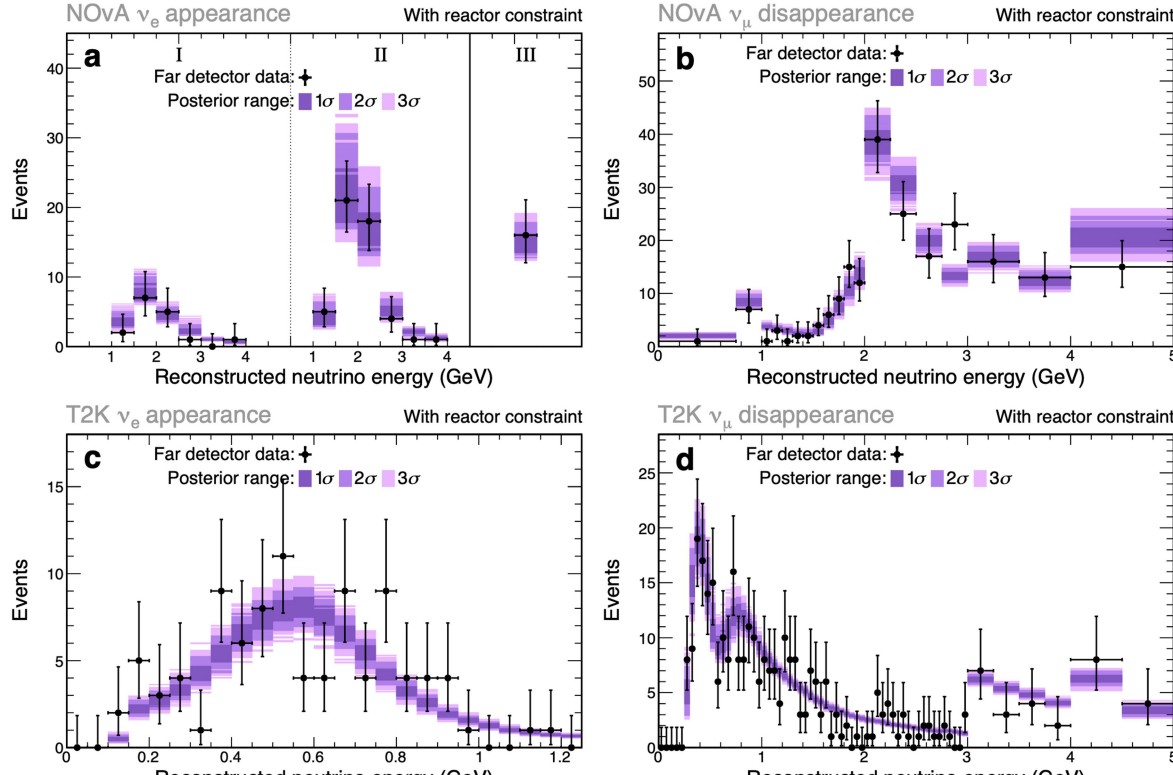

**Extended Data Fig. 5 | NOvA and T2K post-fit spectra.** NOvA (a, b) and T2K (c, d) posterior spectra compared to observed data for the largest $\nu_e$-like (a, c) and $\nu_\mu$-like (b, d) event samples with the beam running enriched in $\nu_\mu$ (as opposed to $\bar\nu_\mu$) extracted from a fit with reactor constraint, marginalized over both mass orderings. The NOvA $\nu_e$-like sample (a) is divided into three subsets as shown here: events with a lower (I) or higher (II) event classification score and events lying near the periphery of the detector (III). Note that T2K also has a $\nu_e$-like sample targeting events with single $\pi$ not shown here.

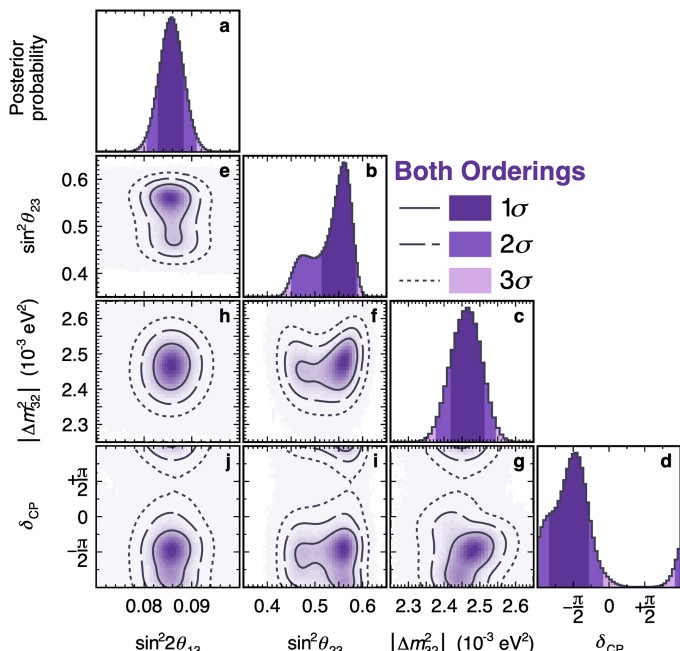

**Extended Data Fig. 6 | Constraints on PMNS oscillation parameters in 1D and 2D for both orderings.** The 1D posterior probability distributions of $\sin^2\theta_{13}$ (a), $\sin^2\theta_{23}$ (b), $|\Delta m^2_{32}|$ (c), $\delta_{CP}$ (d), and corresponding $1\sigma$, $2\sigma$, $3\sigma$ 2D contours $\sin^2\theta_{23} - \sin^2 2\theta_{13}$ (e), $\Delta m^2_{32} - \sin^2\theta_{23}$ (f), $\delta_{CP} - \Delta m^2_{32}$ (g), $\Delta m^2_{32} - \sin^2 2\theta_{13}$ (h), $\delta_{CP} - \sin^2\theta_{23}$ (i), and $\delta_{CP} - \sin^2 2\theta_{13}$ (j) from the joint fit with reactor constraints marginalized over both mass orderings.

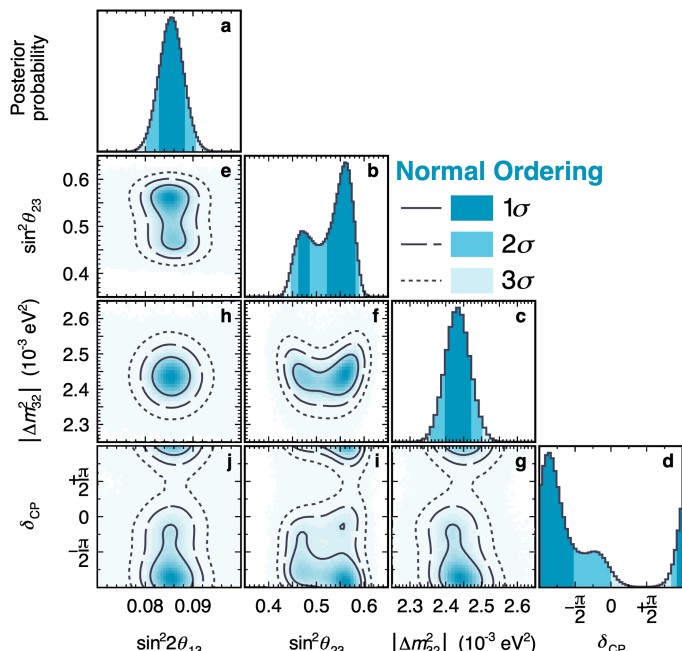

**Extended Data Fig. 7 | Constraints on PMNS oscillation parameters in 1D and 2D for normal ordering.** As in Extended Data Fig. 6, but conditional on the assumption of normal ordering.

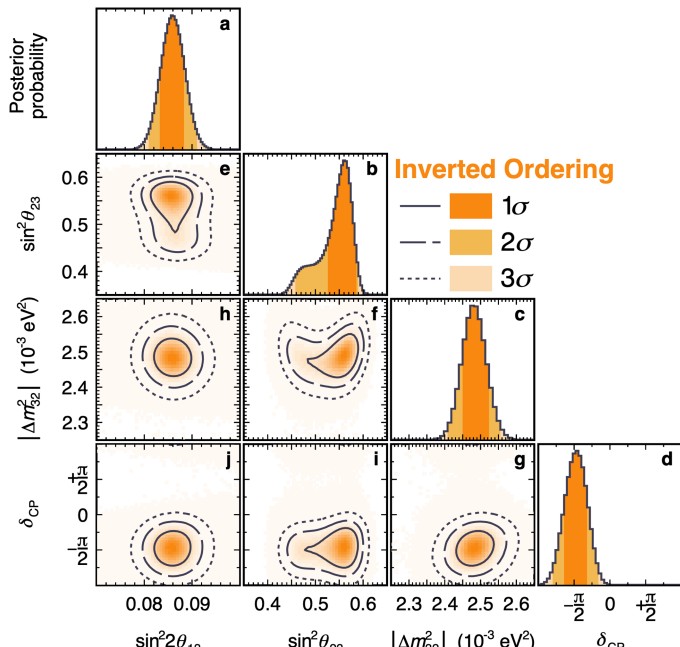

**Extended Data Fig. 8 | Constraints on PMNS oscillation parameters in 1D and 2D for inverted ordering.** As in Extended Data Fig. 6, but conditional on the assumption of inverted ordering.

**a**

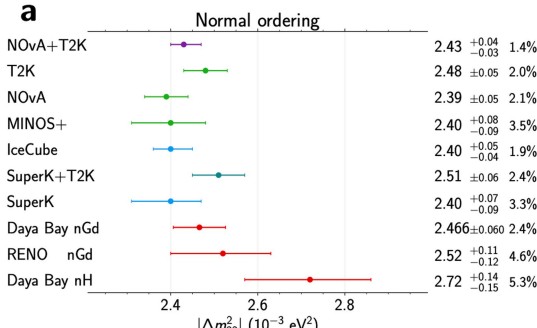

**b**

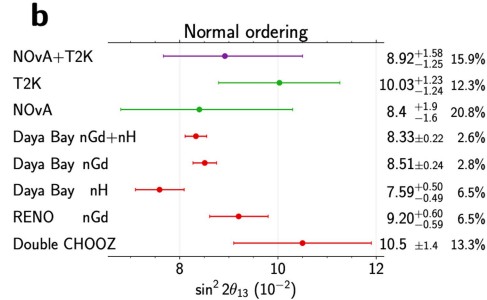

**c**

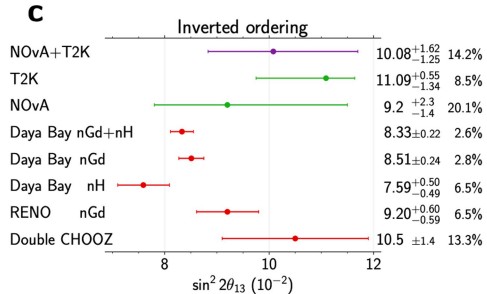

**d**

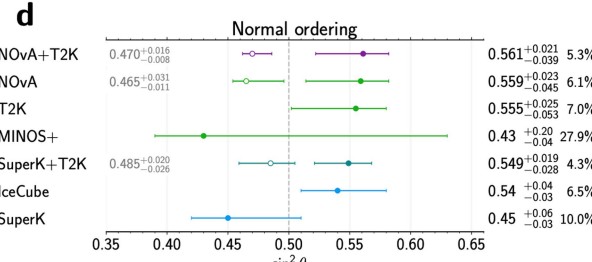

**e**

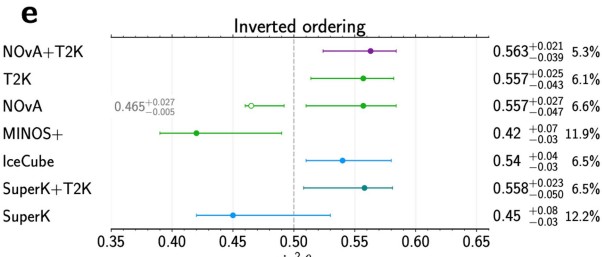

**f**

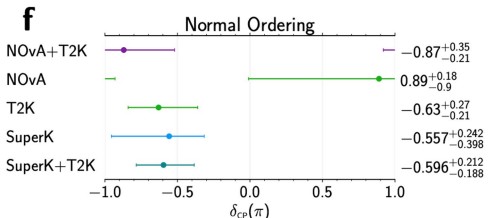

**g**

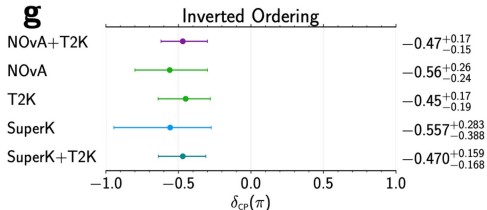

**Extended Data Fig. 9 | Experimental measurements of oscillation parameters.** $|\Delta m_{32}^2|$ assuming normal ordering (a), with sources for the results from top to bottom starting with the second line as follows:[13,14,43–49]. $\sin^2 2\theta_{13}$ assuming normal (b) and inverted (c) ordering, with sources for the results from top to bottom starting with the second line as follows:[13,14,47–49,63]. NOvA+T2K measurement here does not use the reactor constraint. $\sin^2 \theta_{23}$ assuming normal (d) and inverted (e) ordering, with sources for the results from top to bottom starting with the second line as follows:[13,14,43–46]. Open circles denote a local minima position in lower octant. $\delta_{\rm CP}$ assuming normal (f) and inverted (g) ordering, with sources for the results from top to bottom starting with the second line as follows:[13,14,45,46].

**Extended Data Table 1 | Default oscillation parameters for simulation**

|  | T2K-like | NOvA-like | NuFit-like |
|---|---|---|---|
| $\Delta m^2_{32}$ (eV$^2$) | $2.51 \times 10^{-3}$ | $2.41 \times 10^{-3}$ | $-2.45 \times 10^{-3}$ |
| $\sin^2\theta_{23}$ | 0.528 | 0.570 | 0.550 |
| $\delta_{CP}$ | $-0.51\pi$ | $0.83\pi$ | $-0.50\pi$ |

Sets of oscillation parameter values used to generate pseudo-data. For all sets, $\sin^2\theta_{13}$ is $2.18 \times 10^{-2}$, $\Delta m^2_{21}$ is $7.53 \times 10^{-5}$ eV$^2$, and $\sin^2\theta_{12}$ is 0.307.

**Extended Data Table 2 | Posterior predictive *p*-values**

| Channel | Joint *p*-value | | | Subsamples *p*-value | | | | | |
|---|---|---|---|---|---|---|---|---|---|
| | | | | NOvA[a] | | | T2K[b] | | |
| | Both | NO | IO | Both | NO | IO | Both | NO | IO |
| **Rate + Shape** | | | | | | | | | |
| $\nu_e$[c] | 0.62 | 0.53 | 0.69 | 0.90 | 0.83 | 0.95 | 0.19 | 0.18 | $0.20^{(\nu_e)}$ |
| | | | | | | | 0.79 | 0.78 | $0.79^{(\nu_e 1\pi)}$ |
| $\bar{\nu}_e$ | 0.40 | 0.38 | 0.42 | 0.21 | 0.18 | 0.24 | 0.67 | 0.67 | 0.67 |
| $\nu_\mu$ | 0.62 | 0.62 | 0.62 | 0.68 | 0.65 | 0.70 | 0.48 | 0.50 | 0.47 |
| $\bar{\nu}_\mu$ | 0.72 | 0.73 | 0.71 | 0.38 | 0.38 | 0.37 | 0.87 | 0.87 | 0.87 |
| Total | 0.75 | 0.73 | 0.76 | 0.64 | 0.60 | 0.68 | 0.72 | 0.73 | 0.71 |

| Channel | Joint *p*-value | | | Subsamples *p*-value | | | | | |
|---|---|---|---|---|---|---|---|---|---|
| | | | | NOvA | | | T2K | | |
| | Both | NO | IO | Both | NO | IO | Both | NO | IO |
| **Rate** | | | | | | | | | |
| $\nu_e$ | 0.40 | 0.14 | 0.57 | 0.48 | 0.16 | 0.71 | 0.39 | 0.43 | $0.36^{(\nu_e)}$ |
| | | | | | | | 0.11 | 0.12 | $0.11^{(\nu_e 1\pi)}$ |
| $\bar{\nu}_e$ | 0.33 | 0.31 | 0.34 | 0.55 | 0.42 | 0.64 | 0.57 | 0.60 | 0.55 |
| $\nu_\mu$ | 0.15 | 0.17 | 0.14 | 0.24 | 0.23 | 0.25 | 0.11 | 0.10 | 0.12 |
| $\bar{\nu}_\mu$ | 0.93 | 0.94 | 0.93 | 0.90 | 0.89 | 0.90 | 0.72 | 0.72 | 0.72 |
| Total | 0.40 | 0.28 | 0.49 | 0.58 | 0.39 | 0.70 | 0.24 | 0.27 | 0.22 |

Posterior predictive *p*-values extracted from the joint fits, marginalized over both mass orderings, normal mass ordering and inverted mass ordering with the reactor constraint.

a NOvA: NOvA sample by sample from the joint fit.

b T2K: T2K sample by sample from the joint fit, $\nu_e$ and $\nu_e 1\pi$ samples treated independently.

c Joint: $\nu_e$ channel *p*-value includes T2K $\nu_e$, T2K $\nu_e 1\pi$ and NOvA $\nu_e$.

**Extended Data Table 3 | NOvA+T2K measurements of oscillation parameters**

| Parameter | Normal Ordering | Inverted Ordering |
|---|---|---|
| $|\Delta m^2_{32}|$ | $2.43^{+0.04}_{-0.03}$ | $2.48^{+0.04}_{-0.03}$ |
| $\sin^2\theta_{23}$ | $0.561^{+0.021}_{-0.039}$ and $0.470^{+0.016\,a}_{-0.008}$ | $0.563^{+0.021}_{-0.039}$ |
| $\delta_{CP}$ | $-0.87\pi^{+0.35\pi}_{-0.21\pi}$ | $-0.47\pi^{+0.17\pi}_{-0.15\pi}$ |
| $\sin^2 2\theta_{13}$ | $0.0855 \pm 0.0027$ | $0.0859^{+0.0027}_{-0.0025}$ |

Values assume normal and inverted ordering with the reactor constraint applied.

a Local extremum in lower octant of $\sin^2\theta_{23}$.