## [Peer Review file · Nature]

Joint neutrino oscillation analysis from the T2K and NOvA experiments

Corresponding Author: Asher Kaboth

Version 0:

Reviewer comments:

Referee #1

(Remarks to the Author)

The joined combination of both T2K and NOvA results provides 3 sigma intervals on the CP violating phase as well as the largest mass differences, for both normal and inverted ordering. Combination of results from different experiments is not a novelty in the neutrino sector but I value the most the fact that the collaboration themselves analyzed their whole bunch of data. The statistical analysis is very robust and the results are presented in a way to facilitate the comprehension of the main outputs.

While the abstract is clearly written, the conclusions are spread in several paragraphs, which does not allow to have a concise summary. I found this a bit discouraging as the reader has to search among several paragraphs and plots for the relevant informations on the physics results.

In addition, the paper is not "self-consistent" as there is no mention of the transition probabilities used to make the fit. In my vision, this point should be addressed to allow the reader to better understand the correlations among the mixing parameters.

(Remarks on code availability)

It is impossible to check their code as it is not provided; in fact, they state: "Inquiries regarding the data and posteriors used in this result may be directed to the collaborations." and also "The NOvA and T2K collaborations develop and maintain the code used for the simulation of the experimental apparatus and statistical analysis of the raw data used in this result. This code is shared among the collaborations but not publicly distributed. Inquiries regarding the algorithms and methods used in this result may be directed to the collaborations."

Referee #2

(Remarks to the Author)

This paper from the T2K and NovA Collaborations, representing the two currently operating long baseline neutrino oscillation experiments, reports on the much-anticipated joint analysis of their neutrino oscillation data. These results represent decades of effort and are measurements of great importance in particle physics. I believe it is an exceptionally clearly written and substantial paper that deserves publication in Nature.

The paper reports the most precise information on several key parameters in the study of neutrino oscillations. Two large international experiments (DUNE and HyperK), which are now under construction, will continue to pursue these measurements with much greater precision. This combination of data from T2K and Nova represents the best understanding we will have of these parameters until the new experiments begin to operate, and are therefore of great interest in the field.

The paper reports several significant results, including a more precise measurement of the mass difference between two of the neutrino mass states (Δm_{32}^2), and 3 sigma allowed intervals for Δ_{CP} , the CP violating phase. The paper also reports on sensitivity to the so-called neutrino mass ordering (whether there are "two light states and one heavier one or vice versa (inverted ordering)"). Although the data show no preference for either mass ordering, they note that if the mass

ordering is inverted, their results would provide evidence of CP violation in the lepton sector. Therefore, if future experiments find the mass ordering to be inverted, this paper could have already provided evidence for CP violation. It should be noted that the CP violation measurement is of broad interest because of its potential connection to leptogenesis as a mechanism for creating the matter - antimatter asymmetry in the universe.

The paper provides an extremely clear description of the approach to combining the two results. It is particularly thorough on the challenging treatment of potential correlations between systematic uncertainties in the two experiments. The Methods section provides additional details on this procedure. The techniques developed to perform this combined analysis will provide an excellent reference for future combinations of DUNE and HyperK results.

As noted earlier, the paper is excellent and does not require revision before publication. I note two minor points where I would have found additional detail helpful:

- As described in Methods, the two experiments use near detector data in different ways. I would have appreciated some more discussion of whether this difference created specific issues in the combined fit.
- It would be interesting to see an estimate of the potential sensitivity of T2K and Nova combining the full expected data samples. This combination will provide the best understanding of neutrino oscillation parameters until the next generation of experiments (JUNO, DUNE, HyperK) begin to operate.

I noted one typo on line 938: models should be model's.

(Remarks on code availability)

Referee #3

(Remarks to the Author)

This article presents the first combined analysis of the oscillation data by long baseline neutrino experiments T2K and NOvA. It has been eagerly awaited by the community and it is very impactful, especially taking into account the information it provides on leptonic CP violation.

The analysis is highly original and of great significance. By combining the results of the two experiments, it is possible to obtain reliable constraints on the leptonic CP violating phase δ . It is found that, in the case of inverted ordering, the CP-conserving values of δ , 0 and π , are outside the 3-sigma credible intervals. For normal ordering, NOvA and T2K prefer different ranges of δ resulting in a broader allowed interval, that includes its CP-conserving values. There is no strong preference for a mass ordering, leaving this question open to future studies. Additionally, constraints on θ_{23} and Δm^2_{32} are found, the latter one being measured with the smallest experimental uncertainty to date.

The methodology is excellent and described in great detail in the Methods section. There is no questions that the data are of the highest quality with a deep understanding of all systematic errors and an exceedingly careful and advanced treatment of the uncertainties, to which both the Main and Methods sections devote ample space.

The results obtained are very reliable and robust and offer a new level of understanding of neutrino oscillation parameters. We fully support the publication of the article.

Concerning the Main text, we would like to suggest the authors to implement two main suggestions:

- to devote more space (ideally a paragraph) to discuss the importance of the determination of leptonic CPV. Some hints are provided in terms of a possible connection to the baryon asymmetry (in the abstract with Ref.s 7–9 and in the second paragraph of the Main text with just few lines and an additional two references). The current discussion is very brief, lacking any explanation and resulting not clear or informative. At the very least the term “leptogenesis” should be mentioned and a very brief explanation of the mechanism could be provided. Something along the lines of:

It is well known that the baryon asymmetry can be successfully explained in models of neutrino masses

(here quote Ref. 16 but there are other relevant references including: L. Covi, E. Roulet, and F. Vissani, Phys. Lett. B384, 169 (1996), arXiv:hep-ph/9605319 [hep-ph]. A. Pilaftsis, Phys. Rev. D56, 5431 (1997), arXiv:hep-ph/9707235 [hep-ph]. W. Buchmuller and M. Plumacher, Phys. Lett. B431, 354 (1998), arXiv:hep-ph/9710460 [hep-ph].)

both at high scales

(here one could quote S. Davidson and A. Ibarra, Phys. Lett. B535, 25 (2002), arXiv:hep-ph/0202239 [hep-ph]. in which the lower bound on heavy neutrino masses is found)

and much lower ones

(here some of the key articles to cite are E. K. Akhmedov, V. A. Rubakov, and A. Y. Smirnov, Physical Review Letters 81, 1359 (1998), arXiv:hep-ph/9803255. T. Asaka and M. Shaposhnikov, Physics Letters B 620, 17 (2005), arXiv:hep-ph/0505013. M. Drewes, B. Garbrecht, P. Hernandez, M. Kikic, J. Lopez-Pavon, J. Racker, N. Rius, J. Salvado, and D. Teresi, International Journal of Modern Physics A 33, 1842002 (2018), arXiv:1711.02862. and others).

It has been shown that the low energy CP violating phase δ might be fully responsible for the observed baryon asymmetry in specific cases, e.g. for hierarchical very heavy neutrinos (Ref. 7–9) or at the GeV scale (A. Granelli, S. Pascoli and S. T. Petcov, Phys. Rev. D 108 (2023) no.10, L101302 [arXiv:2307.07476 [hep-ph]]),

although a direct connection cannot be inferred due to the presence of other sources of leptonic CP-violation possibly entering leptogenesis in these models. Nevertheless, the observation of leptonic CP violation would be of crucial importance and would demonstrate that one of the Sakharov conditions (Sakharov, A. D. Sov. Phys. Usp. 34, 392–393 (1991)) is satisfied.

- a very detailed discussion of the correlation (or lack of) in the systematic uncertainties is carried out. It occupies nearly three full columns of Main text. Although this is crucial for the reliability of the experimental results and it is of great interest for the experts, its relevance for a broad audience is not clear to us. Indeed, we suggest to reduce significantly this discussion, summarising the key aspects and moving some of it to the Methods section.

Here are some additional comments to be addressed by the authors.

In general, we find that the discussion of the physics motivations provided in the second paragraph of the Main text could be a bit expanded and enriched for clarity. In particular:

- lines 261-262: it would be useful to provide the explicit parameterisation of the PMNS U matrix either here or in the accompanying section;
- lines 275–282: Since the ordering of neutrino masses is of significant importance in the field and one of the major physics goals of the current and future neutrino oscillation programme, we suggest to expand slightly this discussion and to add additional relevant references.
- lines 284: here it is hinted at a possible flavour symmetry in the leptonic sector, but again this discussion would benefit from being extended with a short but meaningful and precise mention of flavour symmetries, both continuous and discrete. We highlight the fact that there are several recent reviews on the topic (in addition to Ref. [15] cited in the article).
- lines 285-287: the same applies here as well.
- lines 337: the discussion of Fig. 1 would benefit from some additional physics explanations. For instance, it is mentioned that the separation for NO vs IO is greater for NOvA than T2K and this is due to the higher energy. Does the baseline play any role? Also, mentioning matter effects and their impact would enhance the clarity of the physics discussion.
- in the caption of Fig. 1, it would be useful to state explicitly the values of the parameters taken or to point to the table/figure in the Methods section where they can be found.
- It is noted that the values of Δm^2_{21} , θ_{12} and θ_{13} are taken from PDG2020. This has been superseded by the current version of the PDG. Although the difference in the values is extremely small and would not make any difference to the results obtained, it would be appropriate to make a comment about this (for instance around line 540), as to clarify that the results are valid and accurate despite the priors from 2020.

We have also minor suggestions on specific aspects of the text.

- line 224: "mass-squared" instead of "mass".
- lines 225-226: The discussion of the definition of normal versus inverted ordering is a bit imprecise as the masses could be have a very mild hierarchy. To minimally change the text, it might suffice to say "whether there are two quasi-degenerate light states and a heavier one (normal ordering) or the two quasi-degenerate ones are the heaviest (inverted ordering)".
- line 228: leptonic mixing instead of flavour mixing.

(Remarks on code availability)

Referee #4

(Remarks to the Author)

The paper describes the first joint analysis of the parameters governing long-baseline neutrino oscillation by the two currently operating long-baseline oscillation experiments, NOvA and T2K. The combination is of particular interest as the two experiments measure oscillations at different baselines and energies and thus are differently sensitive to the impact of the matter effect. The analysis is performed assuming most sources of systematic uncertainty are uncorrelated between the two experiments and a significant fraction of the text is devoted to demonstration that this assumption does not impact the combined results and to the related study of potential bias from alternative model inputs. The primary conclusions of the combined analysis are that the individual experimental results are consistent with each other, that the combined data do not have a statistically significant preference for the normal or inverted mass ordering, and measurements are reported of the "atmospheric parameters" and of the credible interval for δ_{CP} . The measurement of the larger mass splitting has the smallest experimental uncertainty of any to date. In addition to the explicit results, the methodology developed in the paper is of deep interest to the community as future experiments DUNE and HyperK may wish to combine results in a similar fashion.

This analysis has been long anticipated in the neutrino oscillation community and has been performed with great attention to detail. The data analysis is that of the individual experiments which are long established and have published many well-regarded results. The Bayesian statistical analysis techniques employed are similarly well established. I am not familiar with any previous analysis that has performed such an in-depth combination of two oscillation experiments where the full suite of analysis tools from each experiment are used as input to a common fit.

The work to establish the validity of the treatment of correlations (the choice was to assume the uncertainties are uncorrelated between the experiments with only a few exceptions) is complex. The fundamental problem here is that the true underlying physics is certainly correlated in many cases (ie: hadron production model, neutrino interaction model) but because of the complexity of the models and the methods used to constrain them, we don't have a good way to model or quantify the correlation. Given that state of affairs, what the authors have done is quite reasonable - they have laid out

arguments as to why the uncertainties are not correlated in a simple way and have performed tests demonstrating that the treatment (or not) of correlations does not significantly impact the reported results. I suspect that the detailed text in the "Methods" section describing these studies will be a difficult read for non-experts, but it is important to have the details laid out in order for the arguments to be accepted by expert readers. And I think that while some of the justifications may be necessarily a bit "hand waving" the final conclusion that these choices don't move the contours is well demonstrated.

The work to demonstrate the lack of bias from out-of-model variations is similarly very complicated and very important. It is well known in the field that our models of neutrino interactions are incomplete and cannot fully describe world data, so demonstrating an analysis to be robust against reasonable out-of-model variations is critical. The variations that are presented are reasonable and seem to me to do a good job of spanning the phase space covered by the experiments, but I would have appreciated a little more exposition in the text regarding why these three specific variations were chosen and how well they cover the range of possible variations. I understand the choice to show only the results for one of the three examples as plots, but I did find myself wondering about what the other distributions looked like. I wonder if the authors have considered some alternative presentation (perhaps inset panels) that could give a peek at the other distributions with less detail?

Regarding the treatment of priors on oscillation parameters in the fit, I have no objection to the methodology used. However, the discussion in the "Methods" section is extremely brief and does not address the comparison of the prior uniform in δ_{CP} or $\sin(\delta_{CP})$ that is mentioned briefly in the conclusions of the "Main" section and seems interesting even if result is not altered by the choice. I think this is simply a presentation issue that throughout the paper there are brief mentions of the priors such that I was expecting more discussion of them in the Methods section than was provided.

Overall this is an extremely well-written paper reporting interesting methods and results that are of great interest to the community. The careful treatment of systematic uncertainty is critical to establishing the validity of the results and is done well. I have no concerns regarding validity or correctness. I have only a few small suggestions regarding the text and figures:

- 1) "These experiments employ near detectors, sited a short distance from the beam source, that measure a very high neutrino event rate where oscillation effects are negligible." I found the construction of this sentence potentially a bit confusing for non-experts - somehow the way it's constructed doesn't convey that the rate is very high *because* the detectors are close to the source and the clause "where oscillation effects are negligible" seems at first to be related to the high rate rather than again to the location near the source. It's a detail, but I did stumble over the construction when reading.
- 2) In Fig 1, the vertical scales are similar but non-identical which makes it a little confusing to compare the width of the ellipses as intended.
- 3) In Extended Data Fig 5, the "baseline model is shown for comparison" but it doesn't specify the baseline model for what - is it using the baseline model to generate a FD spectrum (so directly comparable to the dots) or is it using the baseline model to generate a CV prediction from the ND (so directly comparable to the purple band)?
- 4) "Under the incorrect ordering assumption, these two techniques are expected to measure different incorrect values [61]." Is the difference not on the order of 1-2%? The way this text is phrased I would have expected something wildly wrong rather than subtly shifted. Perhaps a slight rephrasing would better convey the message?

(Remarks on code availability)

I did not find code at the URLs provided. I am, however, familiar with the size and complexity of the code bases for these kinds of experiments and know it would be impossible to provide a meaningful review by attempting to read code.

Referee #5

(Remarks to the Author)

I co-reviewed this manuscript with one of the reviewers who provided the listed reports.

(Remarks on code availability)

The authors thank the referees for their careful review of the manuscript and their helpful suggestions and comments. We include replies to all comments inline below. Original referee text is colored BLACK and author replies are colored BLUE.

Referee #1 (Remarks to the Author):

The joined combination of both T2K and NOvA results provides 3 sigma intervals on the CP violating phase as well as the largest mass differences, for both normal and inverted ordering. Combination of results from different experiments is not a novelty in the neutrino sector but I value the most the fact that the collaboration themselves analyzed their whole bunch of data. The statistical analysis is very robust and the results are presented in a way to facilitate the comprehension of the main outputs.

While the abstract is clearly written, the conclusions are spread in several paragraphs, which does not allow to have a concise summary. I found this a bit discouraging as the reader has to search among several paragraphs and plots for the relevant informations on the physics results.

We are glad to hear that the summary in the abstract is clear. Indeed, that section is structured to provide the concise enumeration of the key results. At the end of the article, the summary of the results is expanded to provide important context and interpretation. Given that multiple aspects of the neutrino sector are being measured, this naturally takes a few paragraphs to cover, but we feel such length is a helpful complement to the briefer version at the start of the paper. We have also now added Extended Data Table V so that 1D numerical results can be readily obtained.

In addition, the paper is not "self-consistent" as there is no mention of the transition probabilities used to make the fit. In my vision, this point should be addressed to allow the reader to better understand the correlations among the mixing parameters.

During manuscript preparation, we explored options for how much of a review this article should provide, versus focusing on reporting our experimental findings. We opted to use most space for the latter. We did include Figure 1 and its associated discussion since it focuses on the nature of the complementarity between NOvA and T2K (*i.e.*, the complementary in how the CP-violating phase and the mass ordering affect the appearance probabilities.) We opted not to include a full articulation of the transition probabilities since these, when expanded using the typical parameterization and in the presence of matter effects, are very involved and not necessarily insightful without substantial discussion or without approximations that aren't valid in this paper's context. We have included references to the individual experiments' papers and, importantly, a pedagogical review article summarizing the physics and experimental

considerations of long-baseline neutrino oscillations more generally. These sources have the space to help an interested reader learn more about the interplay of the various neutrino sector parameters.

Referee #1 (Remarks on code availability):

It is impossible to check their code as it is not provided; in fact, they state: "Inquiries regarding the data and posteriors used in this result may be directed to the collaborations." and also "The NOvA and T2K collaborations develop and maintain the code used for the simulation of the experimental apparatus and statistical analysis of the raw data used in this result. This code is shared among the collaborations but not publicly distributed. Inquiries regarding the algorithms and methods used in this result may be directed to the collaborations.

The code bases for both experiments are complex and run to many hundreds of thousands of lines of code. We have added an extra clause to clarify this, and welcome inquiries about our methods.

Referee #2 (Remarks to the Author):

This paper from the T2K and NovA Collaborations, representing the two currently operating long baseline neutrino oscillation experiments, reports on the much-anticipated joint analysis of their neutrino oscillation data. These results represent decades of effort and are measurements of great importance in particle physics. I believe it is an exceptionally clearly written and substantial paper that deserves publication in Nature.

The paper reports the most precise information on several key parameters in the study of neutrino oscillations. Two large international experiments (DUNE and HyperK), which are now under construction, will continue to pursue these measurements with much greater precision. This combination of data from T2K and Nova represents the best understanding we will have of these parameters until the new experiments begin to operate, and are therefore of great interest in the field.

The paper reports several significant results, including a more precise measurement of the mass difference between two of the neutrino mass states (Δm_{32}^2), and 3 sigma allowed intervals for Δ_{CP} , the CP violating phase. The paper also reports on sensitivity to the so-called neutrino mass ordering (whether there are "two light states and one heavier one or vice versa (inverted ordering)"). Although the data show no preference for either mass ordering, they note that if the mass ordering is inverted, their

results would provide evidence of CP violation in the lepton sector. Therefore, if future experiments find the mass ordering to be inverted, this paper could have already provided evidence for CP violation. It should be noted that the CP violation measurement is of broad interest because of its potential connection to leptogenesis as a mechanism for creating the matter - antimatter asymmetry in the universe.

The paper provides an extremely clear description of the approach to combining the two results. It is particularly thorough on the challenging treatment of potential correlations between systematic uncertainties in the two experiments. The Methods section provides additional details on this procedure. The techniques developed to perform this combined analysis will provide an excellent reference for future combinations of DUNE and HyperK results.

As noted earlier, the paper is excellent and does not require revision before publication. I note two minor points where I would have found additional detail helpful:

— As described in Methods, the two experiments use near detector data in different ways. I would have appreciated some more discussion of whether this difference created specific issues in the combined fit.

The methods of incorporating near detector data do guide the analysis approaches taken by the individual experiments, but for the combination, these details are abstracted away since we combine things at the likelihood level. Potential connections in model parameters across the two experiments do therefore hold relevance, though, so significant space is given to discussing those aspects. In short, there aren't any issues stemming directly from the different near detector techniques.

— It would be interesting to see an estimate of the potential sensitivity of T2K and Nova combining the full expected data samples. This combination will provide the best understanding of neutrino oscillation parameters until the next generation of experiments (JUNO, DUNE, HyperK) begin to operate.

While we agree this would be of interest to the reader, it would unfortunately be a significant increase in scope. Many of the studies carried out in this work are designed to show that our treatment of systematic correlations and our models themselves (particularly the cross section models) are adequate for the exposures analyzed here. For future larger data sets, these studies would be revisited and may lead to amendments to the analysis methods.

I noted one typo on line 938: models should be model's.

This has been fixed.

Referee #3 (Remarks to the Author):

This article presents the first combined analysis of the oscillation data by long baseline neutrino experiments T2K and NOvA. It has been eagerly awaited by the community and it is very impactful, especially taking into account the information it provides on leptonic CP violation.

The analysis is highly original and of great significance. By combining the results of the two experiments, it is possible to obtain reliable constraints on the leptonic CP violating phase δ . It is found that, in the case of inverted ordering, the CP-conserving values of δ , 0 and π , are outside the 3-sigma credible intervals. For normal ordering, NOvA and T2K prefer different ranges of δ resulting in a broader allowed interval, that includes its CP-conserving values. There is no strong preference for a mass ordering, leaving this question open to future studies. Additionally, constraints on θ_{23} and Δm^2_{32} are found, the latter one being measured with the smallest experimental uncertainty to date.

The methodology is excellent and described in great detail in the Methods section. There is no questions that the data are of the highest quality with a deep understanding of all systematic errors and an exceedingly careful and advanced treatment of the uncertainties, to which both the Main and Methods sections devote ample space.

The results obtained are very reliable and robust and offer a new level of understanding of neutrino oscillation parameters. We fully support the publication of the article.

Concerning the Main text, we would like to suggest the authors to implement two main suggestions:

- to devote more space (ideally a paragraph) to discuss the importance of the determination of leptonic CPV. Some hints are provided in terms of a possible connection to the baryon asymmetry (in the abstract with Ref.s 7–9 and in the second paragraph of the Main text with just few lines and an additional two references). The current discussion is very brief, lacking any explanation and resulting not clear or informative. At the very least the term “leptogenesis” should be mentioned and a very brief explanation of the mechanism could be provided. Something along the lines of:

It is well known that the baryon asymmetry can be successfully explained in models of neutrino masses (here quote Ref. 16 but there are other relevant references including: L. Covi, E. Roulet, and F. Vissani, Phys. Lett. B384, 169 (1996), arXiv:hep-ph/9605319 [hep-ph]. A. Pilaftsis, Phys. Rev. D56, 5431 (1997), arXiv:hep-ph/9707235 [hep-ph]. W. Buchmuller and M. Plumacher, Phys. Lett. B431, 354 (1998), arXiv:hep-ph/9710460 [hep-ph].)

both at high scales

(here one could quote S. Davidson and A. Ibarra, Phys. Lett. B535, 25 (2002), arXiv:hep-ph/0202239 [hep-ph]. in which the lower bound on heavy neutrino masses is found)

and much lower ones

(here some of the key articles to cite are E. K. Akhmedov, V. A. Rubakov, and A. Y. Smirnov, Physical Review Letters 81, 1359 (1998), arXiv:hep-ph/9803255. T. Asaka and M. Shaposhnikov, Physics Letters B 620, 17 (2005), arXiv:hep-ph/0505013. M. Drewes, B. Garbrecht, P. Hernandez, M. Kekic, J. Lopez-Pavon, J. Racker, N. Rius, J. Salvado, and D. Teresi, International Journal of Modern Physics A 33, 1842002 (2018), arXiv:1711.02862. and others).

It has been shown that the low energy CP violating phase δ might be fully responsible for the observed baryon asymmetry in specific cases, e.g. for hierarchical very heavy neutrinos (Ref. 7–9) or at the GeV scale (A. Granelli, S. Pascoli and S. T. Petcov, Phys. Rev. D 108 (2023) no.10, L101302 [arXiv:2307.07476 [hep-ph]].),

although a direct connection cannot be inferred due to the presence of other sources of leptonic CP-violation possibly entering leptogenesis in these models. Nevertheless, the observation of leptonic CP violation would be of crucial importance and would demonstrate that one of the Sakharov conditions (Sakharov, A. D. Sov. Phys. Usp. 34, 392–393 (1991)) is satisfied.

We thank the referee for the excellent list of potential references. Given the breadth of work in the area of leptogenesis and the constraints on citation count, it is a challenge to itemize most articles. For this reason, we have relied heavily on well-regarded review articles in our citation list, with a balance of primary sources and review articles that seems to align well with other “letter”-length experimental papers on neutrino oscillations. Your comments did remind us of another nice review that, in turn, cites many of the suggestions above, so we have added that reference – W. Buchmuller, R. D. Peccei, and T. Yanagida, “Leptogenesis as the origin of matter”, Ann. Rev. Nucl. Part. Sci. 55, 311 (2005), arXiv:hep-ph/0502169. We have also unified the two separate citation blocks as you suggest so that M. Fukugita and T. Yanagida (1986) is not separated from the rest. Regarding expanding the CPv text, we refer you to a relevant reply below.

- a very detailed discussion of the correlation (or lack of) in the systematic uncertainties is carried out. It occupies nearly three full columns of Main text. Although this is crucial for the reliability of the experimental results and it is of great interest for the experts, its relevance for a broad audience is not clear to us. Indeed, we suggest to reduce significantly this discussion, summarising the key aspects and moving some of it to the Methods section.

During the development of this joint analysis, the broader neutrino community showed significant interest in how we would deal with the challenges of systematic correlations. Initial versions of the manuscript indeed had even more detail here, but for precisely the reasons you note, we compressed the discussion to a level that we felt aligned with the neutrino community's expressed interest (if perhaps not the interest of a much wider audience). As the field looks toward the next generation of experiments – and future combinations of their data sets – these techniques will only grow in relevance. We have opted not to reduce this section.

Here are some additional comments to be addressed by the authors.

In general, we find that the discussion of the physics motivations provided in the second paragraph of the Main text could be a bit expanded and enriched for clarity. In particular:

- lines 261-262: it would be useful to provide the explicit parameterisation of the PMNS U matrix either here or in the accompanying section;

- lines 275–282: Since the ordering of neutrino masses is of significant importance in the field and one of the major physics goals of the current and future neutrino oscillation programme, we suggest to expand slightly this discussion and to add additional relevant references.

- lines 284: here it is hinted at a possible flavour symmetry in the leptonic sector, but again this discussion would benefit from being extended with a short but meaningful and precise mention of flavour symmetries, both continuous and discrete. We highlight the fact that there are several recent reviews on the topic (in addition to Ref. [15] cited in the article).

- lines 285-287: the same applies here as well.

These items and the earlier CPv text suggestions connect to our previous reply. Since this manuscript reports measurements on multiple parameters in the neutrino sector, additional pedagogical or phenomenological discussion for each aspect would lead to significant space increase while not necessarily adding enough information to cross a threshold of usefulness to readers less familiar with the field. We have included references to the individual experiments' papers and, importantly, a pedagogical review article summarizing the physics and experimental considerations of long-baseline neutrino oscillations more generally.

- lines 337: the discussion of Fig. 1 would benefit from some additional physics explanations. For instance, it is mentioned that the separation for NO vs IO is greater for NOvA than T2K and this is due to the higher energy. Does the baseline play any role? Also, mentioning matter effects and their impact would enhance the clarity of the physics discussion.

Matter effects as neutrinos pass through the earth's crust form the main effect that, in long-baseline experiments, permits disambiguation of the mass orderings. The magnitude of matter effects is directly dependent on neutrino energy; the baseline only comes in to ensure that an experiment's far detector is located at the first oscillation maximum. For the reasons noted in the reply above, we have not elaborated on these nuances in the text.

- in the caption of Fig. 1, it would be useful to state explicitly the values of the parameters taken or to point to the table/figure in the Methods section where they can be found.

We have added a table with the explicit parameter values (Extended Data Table V).

- It is noted that the values of Δm^2_{21} , θ_{12} and θ_{13} are taken from PDG2020. This has been superseded by the current version of the PDG. Although the difference in the values is extremely small and would not make any difference to the results obtained, it would be appropriate to make a comment about this (for instance around line 540), as to clarify that the results are valid and accurate despite the priors from 2020.

The 2020 version of the PDG was used as it was the current version at the time of the original analyses, and updates since then do not change any conclusions. We have added a comment to this effect in the Methods section pertaining to the priors.

We have also minor suggestions on specific aspects of the text.

- line 224: "mass-squared" instead of "mass".

We have made this change.

- lines 225-226: The discussion of the definition of normal versus inverted ordering is a bit imprecise as the masses could be have a very mild hierarchy. To minimally change the text, it might suffice to say "whether there are two quasi-degenerate light states and a heavier one (normal ordering) or the two quasi-degenerate ones are the heaviest (inverted ordering)".

We tried a few modifications to capture this point, but all require a trade-off between accessibility and precision. Given that this is the abstract aimed at the widest possible audience, we have opted to keep the original wording.

- line 228: leptonic mixing instead of flavour mixing.

We intend here to describe that the mass states are mixtures of the flavor states and vice versa and that we will probe the structure of this mixing rather than describing two different aspects. We've changed it to "mass and flavor mixing" to make this clearer and to align better with language used elsewhere in the literature.

Referee #4 (Remarks to the Author):

The paper describes the first joint analysis of the parameters governing long-baseline neutrino oscillation by the two currently operating long-baseline oscillation experiments, NOvA and T2K. The combination is of particular interest as the two experiments measure oscillations at different baselines and energies and thus are differently sensitive to the impact of the matter effect. The analysis is performed assuming most sources of systematic uncertainty are uncorrelated between the two experiments and a significant fraction of the text is devoted to demonstration that this assumption does not impact the combined results and to the related study of potential bias from alternative model inputs. The primary conclusions of the combined analysis are that the individual experimental results are consistent with each other, that the combined data do not have a statistically significant preference for the normal or inverted mass ordering, and measurements are reported of the "atmospheric parameters" and of the credible interval for δ_{CP} . The measurement of the larger mass splitting has the smallest experimental uncertainty of any to date. In addition to the explicit results, the methodology developed in the paper is of deep interest to the community as future experiments DUNE and HyperK may wish to combine results in a similar fashion.

This analysis has been long anticipated in the neutrino oscillation community and has been performed with great attention to detail. The data analysis is that of the individual experiments which are long established and have published many well-regarded results. The Bayesian statistical analysis techniques employed are similarly well established. I am not familiar with any previous analysis that has performed such an in-depth combination of two oscillation experiments where the full suite of analysis tools from each experiment are used as input to a common fit.

The work to establish the validity of the treatment of correlations (the choice was to assume the uncertainties are uncorrelated between the experiments with only a few exceptions) is complex. The fundamental problem here is that the true underlying

physics is certainly correlated in many cases (ie: hadron production model, neutrino interaction model) but because of the complexity of the models and the methods used to constrain them, we don't have a good way to model or quantify the correlation. Given that state of affairs, what the authors have done is quite reasonable - they have laid out arguments as to why the uncertainties are not correlated in a simple way and have performed tests demonstrating that the treatment (or not) of correlations does not significantly impact the reported results. I suspect that the detailed text in the "Methods" section describing these studies will be a difficult read for non-experts, but it is important to have the details laid out in order for the arguments to be accepted by expert readers. And I think that while some of the justifications may be necessarily a bit "hand waving" the final conclusion that these choices don't move the contours is well demonstrated.

The work to demonstrate the lack of bias from out-of-model variations is similarly very complicated and very important. It is well known in the field that our models of neutrino interactions are incomplete and cannot fully describe world data, so demonstrating an analysis to be robust against reasonable out-of-model variations is critical. The variations that are presented are reasonable and seem to me to do a good job of spanning the phase space covered by the experiments, but I would have appreciated a little more exposition in the text regarding why these three specific variations were chosen and how well they cover the range of possible variations. I understand the choice to show only the results for one of the three examples as plots, but I did find myself wondering about what the other distributions looked like. I wonder if the authors have considered some alternative presentation (perhaps inset panels) that could give a peek at the other distributions with less detail?

The main body had included text later on about the three studies chosen being the ones most impactful in the T2K-only analysis. This was the driving motivation behind picking these three as there was no reason for the lower impact studies to change their order of impact in the joint fit. We have moved the relevant text directly next to the introduction of the alternative model section for better clarity. We explored including more of the results in the extended data sections but it is a very large number of plots and there was a desire to balance the space given to each aspect of the analysis. The figures shown are very representative of the other cases, and all results show insignificant systematic effects.

Regarding the treatment of priors on oscillation parameters in the fit, I have no objection to the methodology used. However, the discussion in the "Methods" section is extremely brief and does not address the comparison of the prior uniform in δ_{CP} or $\sin(\delta_{CP})$ that is mentioned briefly in the conclusions of the "Main" section and seems interesting even if result is not altered by the choice. I think this is simply a

presentation issue that throughout the paper there are brief mentions of the priors such that I was expecting more discussion of them in the Methods section than was provided.

We have added new text here noting that the flat-in-sin(δ_{cp}) prior did not change any conclusions compared to the flat-in- δ_{cp} prior.

Overall this is an extremely well-written paper reporting interesting methods and results that are of great interest to the community. The careful treatment of systematic uncertainty is critical to establishing the validity of the results and is done well. I have no concerns regarding validity or correctness. I have only a few small suggestions regarding the text and figures:

1) "These experiments employ near detectors, sited a short distance from the beam source, that measure a very high neutrino event rate where oscillation effects are negligible." I found the construction of this sentence potentially a bit confusing for non-experts - somehow the way it's constructed doesn't convey that the rate is very high *because* the detectors are close to the source and the clause "where oscillation effects are negligible" seems at first to be related to the high rate rather than again to the location near the source. It's a detail, but I did stumble over the construction when reading.

We agree and have rephrased the sentence.

2) In Fig 1, the vertical scales are similar but non-identical which makes it a little confusing to compare the width of the ellipses as intended.

We have updated the figure to ensure identical vertical scales.

3) In Extended Data Fig 5, the "baseline model is shown for comparison" but it doesn't specify the baseline model for what - is it using the baseline model to generate a FD spectrum (so directly comparable to the dots) or is it using the baseline model to generate a CV prediction from the ND (so directly comparable to the purple band)?

We have changed the language in the figure and caption to match the main body text saying that is the central value of our nominal model.

4) "Under the incorrect ordering assumption, these two techniques are expected to measure different incorrect values [61]." Is the difference not on the order of 1-2%? The way this text is phrased I would have expected something wildly wrong rather than subtly shifted. Perhaps a slight rephrasing would better convey the message?

We have adjusted the phrasing, added a numerical range for the effect, and updated the citation to the much more recent work by Parke and Funchal on these points.

Referee #4 (Remarks on code availability):

I did not find code at the URLs provided. I am, however, familiar with the size and complexity of the code bases for these kinds of experiments and know it would be impossible to provide a meaningful review by attempting to read code.

Referee #5 (Remarks to the Author):

I co-reviewed this manuscript with one of the reviewers who provided the listed reports.